# Reliable Learning of Halfspaces under Gaussian Marginals

**Ilias Diakonikolas**
University of Wisconsin-Madison
ilias@cs.wisc.edu

**Lisheng Ren**
University of Wisconsin-Madison
lren29@wisc.edu

**Nikos Zarifis**
University of Wisconsin-Madison
zarifis@wisc.edu

## Abstract

We study the problem of PAC learning halfspaces in the reliable agnostic model of Kalai et al. (2012). The reliable PAC model captures learning scenarios where one type of error is costlier than the others. Our main positive result is a new algorithm for reliable learning of Gaussian halfspaces on $\mathbb{R}^d$ with sample and computational complexity

$$d^{O(\log(\min\{1/\alpha, 1/\epsilon\}))} \min(2^{\log(1/\epsilon)^{O(\log(1/\alpha))}}, 2^{\text{poly}(1/\epsilon)}) \,,$$

where $\epsilon$ is the excess error and $\alpha$ is the bias of the optimal halfspace. We complement our upper bound with a Statistical Query lower bound suggesting that the $d^{\Omega(\log(1/\alpha))}$ dependence is best possible. Conceptually, our results imply a strong computational separation between reliable agnostic learning and standard agnostic learning of halfspaces in the Gaussian setting.

## 1 Introduction

Halfspaces (or Linear Threshold Functions) is the class of functions $f : \mathbb{R}^d \to \{\pm 1\}$ of the form $f(\mathbf{x}) = \text{sign}(\langle \mathbf{w}, \mathbf{x} \rangle - t)$, where $\mathbf{w} \in \mathbb{R}^d$ is called the weight vector and $t$ is called the threshold. The problem of learning halfspaces is one of the classical and most well-studied problems in machine learning—going back to the Perceptron algorithm [Ros58]—and has had great impact on many other influential techniques, including SVMs [Vap98] and AdaBoost [FS97].

Here we focus on learning halfspaces from random labeled examples. The computational complexity of this task crucially depends on the choice of the underlying model. For example, in the realizable PAC model (i.e., with clean labels), the problem is known to be efficiently solvable (see, e.g., [MT94]) via a reduction to linear programming. Unfortunately, this method is quite fragile and breaks down in the presence of noisy labels. In the noisy setting, the computational complexity of the problem depends on the choice of noise model and distributional assumptions. In this work, we study the problem of distribution-specific PAC learning of halfspaces, with respect to Gaussian marginals, in the reliable agnostic model of [KKM12]. Formally, we have the following definition.

**Definition 1.1** ((Positive) Reliable Learning of Gaussian Halfspaces). *Let $\mathcal{H}_d$ be the class of halfspaces on $\mathbb{R}^d$. Given $0 < \epsilon < 1$ and i.i.d. samples $(\mathbf{x}, y)$ from a distribution $D$ supported on $\mathbb{R}^d \times \{\pm 1\}$, where the marginal $D_{\mathbf{x}}$ is the standard Gaussian $\mathcal{N}(\mathbf{0}, \mathbf{I})$, we say that an algorithm reliably learns $\mathcal{H}_d$ to error $\epsilon$ if the algorithm with high probability outputs a hypothesis $h : \mathbb{R}^d \to \{\pm 1\}$ such that $\mathbf{Pr}_{(\mathbf{x}, y) \sim D}[h(\mathbf{x}) = 1 \wedge y = -1] \leq \epsilon$ and $\mathbf{Pr}_{(\mathbf{x}, y) \sim D}[h(\mathbf{x}) = -1 \wedge y = 1] \leq \text{OPT} + \epsilon$,*

38th Conference on Neural Information Processing Systems (NeurIPS 2024).

*where* $\mathrm{OPT} \overset{\text{def}}{=} \min_{f \in \mathcal{G}(D)} \mathbf{Pr}_{(\mathbf{x},y) \sim D}[f(\mathbf{x}) = -1 \wedge y = 1]$, *with*

$$\mathcal{G}(D) = \{f \in \mathcal{H}_d : \mathbf{Pr}_{(\mathbf{x},y) \sim D}[f(\mathbf{x}) = 1 \wedge y = -1] = 0\} \cup \{f(\mathbf{x}) = -1\} .$$

*We say that $f = \operatorname{argmin}_{f \in \mathcal{G}(D)} \mathbf{Pr}_{(\mathbf{x},y) \sim D}[f(\mathbf{x}) = -1 \wedge y = 1]$ is the optimal halfspace on distribution $D$ and for the conditions above that hypothesis $h$ satisfies, we say that $h$ is $\epsilon$-reliable with respect to $\mathcal{H}_d$ on distribution $D$.*

In other words, a (positive) reliable learner makes almost no false positive errors, while also maintaining the best possible false negative error—as compared to any hypothesis with no false positive errors. Note that if there is no hypothesis (in the target class) that makes no false positive errors, then essentially we have no requirement for the false negative error of the returned hypothesis. The algorithm can then simply return the $-1$ constant hypothesis.

The reliable agnostic PAC model was introduced by [KKM12] as a one-sided analogue of the classical agnostic model [Hau92, KSS94], and has since been extensively studied in a range of works; see, e.g., [KKM12, KT14, GKKT17, DJ19, GKKM20, KK21]. The underlying motivation for this definition comes from learning situations where mistakes of one type (e.g., false positive) should be avoided at all costs. Typical examples include spam detection—where incorrectly detecting spam emails is much less costly than mislabeling an important email as spam—and detecting network failures—where false negatives may be particularly harmful. Such scenarios motivate the study of reliable learning, which can be viewed as minimizing a loss function for different costs for false positive and false negative errors (see, e.g., [Dom99, Elk01]). In a historical context, reliable learning is related to the Neyman-Pearson criterion [NP33] in hypothesis testing, which shows that the optimal strategy to minimize one type of error subject to another type of error being bounded is to threshold the likelihood ratio function. More recently, reliable learning has been shown [KK21] to be equivalent to the PQ-learning model [GKKM20]—a recently introduced learning model motivated by covariance shift.

The algorithmic task of agnostic reliable learning can be quite challenging. While reliable learning can be viewed as minimizing a loss function with different cost for false positive and false negative error, in general, such a loss function will result in a non-convex optimization problem. As pointed out in [KKM12], distribution-specific reliable learning can be efficiently reduced to distribution-specific agnostic learning (such reduction preserves the marginal distribution). A natural question, serving as a motivation for this work, is whether reliable learning is qualitatively easier computationally—for natural concept classes and halfspaces in particular.

Before we state our contributions, we briefly summarize prior work on agnostically learning Gaussian halfspaces. Recall that, in agnostic learning, the goal is to output a hypothesis with error $\mathrm{OPT} + \epsilon$, where OPT is the optimal 0-1 error within the target class. Prior work has essentially characterized the computational complexity of agnostically learning Gaussian halfspaces. Specifically, it is known that the $L_1$ polynomial regression algorithm of [KKMS08] is an agnostic learner with complexity $d^{O(1/\epsilon^2)}$ [DGJ$^+$09, DKN10]. Moreover, there is strong evidence that this complexity upper bound is tight, both in the Statistical Query (SQ) model [DKZ20, GGK20, DKPZ21] and under plausible cryptographic assumptions [DKR23, Tie23]. It is worth noting that the aforementioned hardness results hold even for the subclass of homogeneous halfspaces.

Given the aforementioned reduction of reliable learning to agnostic learning [KKM12], one can use $L_1$-regression as a reliable agnostic learner for Gaussian halfspaces, leading to complexity $d^{O(1/\epsilon^2)}$. Prior to this work, this was the best (and only known) reliable halfspace learner. Given the fundamental nature of this problem, it is natural to ask if one can do better for reliable learning.

> *Is it possible to develop faster algorithms for reliably learning Gaussian halfspaces compared to agnostic learning?*

In this work, we provide an affirmative answer to this question.

To formally state our main result, we need the notion of the bias of a Boolean function.

**Definition 1.2** (Bias of Boolean Function). *We define the bias $\alpha \in [0, 1/2]$ of $h : \mathbb{R}^d \to \{\pm 1\}$ under the Gaussian distribution as $\alpha = \alpha(h) \overset{\text{def}}{=} \min(\mathbf{Pr}_{\mathbf{x} \sim \mathcal{N}_d}[h(\mathbf{x}) = 1], \mathbf{Pr}_{\mathbf{x} \sim \mathcal{N}_d}[h(\mathbf{x}) = -1])$.*

The main result of this paper is encapsulated in the following theorem:

**Theorem 1.3** (Main Result)**.** *Let $D$ be a joint distribution of $(\mathbf{x}, y)$ supported on $\mathbb{R}^d \times \{\pm 1\}$ with marginal $D_{\mathbf{x}} = \mathcal{N}_d$ and let $\alpha$ be the bias of the optimal halfspace on distribution $D$ (with respect to Definition 1.1). There is an algorithm that uses $N = d^{O(\log(\min\{1/\alpha, 1/\epsilon\}))} \min\left(2^{\log(1/\epsilon)^{O(\log(1/\alpha))}}, 2^{\text{poly}(1/\epsilon)}\right)$ many samples from $D$, runs in $\text{poly}(N, d)$ time, and with high probability returns a hypothesis $h(\mathbf{x}) = \text{sign}(\langle \mathbf{w}, \mathbf{x} \rangle - t)$ that is $\epsilon$-reliable with respect to $\mathcal{H}_d$. Moreover, for $\epsilon < \alpha/2$, any SQ algorithm for the problem requires complexity $d^{\Omega(\log(1/\alpha))}$.*

For more detailed theorem statements of our upper and lower bounds, see Appendix B for the algorithmic result and Appendix C for the SQ hardness result.

Theorem 1.3 gives a significantly more efficient algorithm (in terms of both sample complexity and computational complexity) for reliably learning Gaussian halfspaces, compared to agnostic learning. Specifically, as long as $\alpha > 0$ is a universal constant, the overall complexity is polynomial in $d$ and quasi-polynomial in $1/\epsilon$—as opposed to $d^{\text{poly}(1/\epsilon)}$ in the agnostic setting. While the complexity of our algorithm (namely, the multiplicative factor that is independent of $d$) increases as the optimal halfspace becomes more biased, it is always bounded above by the complexity of agnostic learning.

Finally, we note that our algorithm also applies to the fully reliable learning model [KKM12], since one can easily reduce the fully reliable learning to positive reliable and negative reliable learning (as observed in [KKM12]).

## 1.1 Overview of Techniques

Instead of directly considering the optimal halfspace for a distribution $D$ (as defined in Definition 1.1), we introduce the following definition as a relaxation.

**Definition 1.4** (Reliability Condition)**.** *We say that a distribution $D$ supported on $X \times \{\pm 1\}$ satisfies the reliability condition with respect to $f : X \mapsto \{\pm 1\}$ if $\mathbf{Pr}_{(\mathbf{x}, y) \sim D}[f(\mathbf{x}) = +1 \wedge y = -1] = 0$ .*

Notice that $h$ being $\epsilon$-reliable on distribution $D$ is equivalent to $h$ having better false negative error compared with any $f$ such that $D$ satisfies the reliability condition with respect to $f$ (instead of compared with the optimal halfspace). At the same time, given any fixed $f$, such a definition allows the adversary to arbitrarily corrupt negative labels, thus allowing for more manageable analysis.

We start with a brief overview of our SQ lower bound construction, followed by the high-level idea enabling our nearly matching algorithm.

**SQ Lower Bound**    As follows from Definition 1.4 and the subsequent discussion, the adversary in reliable learning can arbitrarily corrupt the negative labels. We leverage this idea to construct an instance where the corruption level suffices to match many moments, meaning that labels $y$ are uncorrelated with any polynomial of degree at most $\log(1/\alpha)$. To prove this, we construct an (infinite) feasibility Linear Program (LP) with the following properties: if the LP is feasible, there exists an instance for which no low-degree polynomial correlates with the labels; see Lemma 3.3. To show the existence of such an instance, we leverage a technique from [DKPZ21] that exploits (an infinite generalization of) LP duality. Specifically, we show that it is possible to add noise to the labels whose uncorrupted value is negative so that all low-degree moments are uncorrelated with the observed labels $y$. This implies that no algorithm that relies on low-degree polynomials can succeed for reliable learning. This allows us to show that SQ algorithms fail if they do not use queries with tolerance at most $1/d^{\Omega(\log(1/\alpha))}$ or exponentially many queries.

**Reliable Learning Algorithm**    As discussed in the previous paragraph, an adversary can arbitrarily corrupt the negative labels which can make various algorithmic approaches fail. In more detail, the adversary can corrupt $\Omega(\log(1/\alpha))$ moments; that is, any SQ algorithm needs to rely on higher-order moment information. One of the main difficulties of this setting is that there may exist weight vectors $\mathbf{w}, \mathbf{w}'$ with $\|\mathbf{w} - \mathbf{w}'\|_2 \geq \Omega(1)$ while at the same time the 0-1 error of $\mathbf{w}$ and $\mathbf{w}'$ is nearly the same. This might suggest that no approach can verify that the algorithm decreases the angle between the current weight vector and the optimal vector. To overcome this obstacle, we develop an algorithm that performs a random walk over a "low-dimensional" subspace. To establish the correctness of our

algorithm, we prove that at some point during its execution, the algorithm will find a weight vector sufficiently close to the optimal vector.

To show that such a low-dimensional subspace exists, we proceed as follows: We first prove that there exists a non-trivial polynomial $p$ of degree $O(\log(1/\alpha))$ that correlates with the negative labels (by the term nontrivial, we are referring to the requirement that we cannot use the constant polynomial, as this trivially correlates with the labels); see Claim B.5. This implies that the low-degree moment tensors, i.e., $\mathbf{E}_{(\mathbf{x},y)\sim\mathcal{D}}[\mathbb{1}\{y = -1\}\mathbf{x}^{\otimes k}]$ for some $k$ between 1 and $O(\log(1/\alpha))$, correlate non-trivially with $(\mathbf{w}^*)^{\otimes k}$, where $\mathbf{w}^*$ is the target weight vector. Thus, we can use these moment tensors to construct a low-dimensional subspace. We leverage the structure of the problem, namely that the (observed) negative labels are uncorrupted, to show that the correlation is in fact almost constant. As a consequence, this allows us to construct a subspace whose dimension depends only on the degree of the polynomial $p$ (which is $O(\log(1/\alpha))$) and the desired excess error $\epsilon$.

We then proceed as follows: in each round, as long as the current solution $\mathbf{w}$ is not optimal, i.e., there are negative samples on the region $\{\mathbf{x} \in \mathbb{R}^d : \mathbf{w} \cdot \mathbf{x} - t \geq 0\}$, our algorithm conditions on a thin strip, projects the points $\mathbf{x}$ on the orthogonal complement of $\mathbf{w}$, and reapplies the above structure result. This leads (with constant probability) to an increase in the correlation between $\mathbf{w}$ and $\mathbf{w}^*$. Assuming that the correlation is increased by $\beta$ in each round with probability at least $1/3$, we roughly need at most $1/\beta$ successful updates. Therefore, if we run our algorithm for $3^{1/\beta}$ steps, it is guaranteed that with probability at least $1/3$ we will find a vector almost parallel to $\mathbf{w}^*$.

**Prior Algorithmic Techniques**   Roughly speaking, prior techniques for reliable learning relied on some variant of $L_1$-regression. In that sense, our algorithmic approach departs from a direct polynomial approximation. Concretely, for distribution-free reliable learning, the algorithm from [KT14] uses a one-sided variant of $L_1$ regression—where instead of approximating the target function in $L_1$ norm, they use the hinge loss instead. For distribution-specific (and Gaussian in particular) reliable learning of halfspaces, the only previous approach uses the reduction of [KKM12] to agnostic learning. While our algorithm also leverages polynomial approximation, our approach employs significantly different ideas and we believe it provides a novel perspective for this problem.

**Technical Comparison with Prior Work**   Our algorithm shares similarities with the algorithm of [DKK$^+$22] for learning Gaussian halfspaces with Massart noise (a weaker semi-random noise model). Both algorithms perform a random walk in order to converge to the target halfpace. Having said so, our algorithm is fundamentally different than theirs for the following reasons. The algorithm of [DKK$^+$22] partitions $\mathbb{R}^d$ into sufficiently angles and searches in each one of them for a direction to update the current hypothesis at random. Our algorithm instead conditions on $y = -1$, which are the points that are uncorrupted, and uses the guarantee (that we establish) that the first $\Omega(\log(1/\alpha))$ moments of the distribution (conditioned on $y = -1$) correlate with the unknown optimal hypothesis. This leads to an algorithm with significantly faster runtime.

Our SQ lower bound leverages techniques from [DKPZ21]. Essentially, we formulate a linear program to construct a noise function that makes all low-degree polynomials uncorrelated with the labels $y$. This condition suffices to show that no SQ algorithm can solve the problem without relying on higher moments. To prove the existence of such a noise function, we use (a generalization of) LP duality of an infinite LP, which provides the necessary conditions for the existence of such a function.

## 1.2   Related Work

This work is part of the broader research program of characterizing the efficient learnability of natural concept classes with respect to challenging noise models. The complexity of this general learning task depends on the underlying class, the distributional assumptions and the choice of noise model. A long line of prior work has made substantial progress in this direction for the class of halfspaces under Gaussians (and other natural distributions), and for both semi-random [ABHU15, DKTZ20a, ZSA20, DKTZ20b, DKK$^+$20, DKK$^+$21b, DKK$^+$22] and adversarial noise [KKMS08, KOS08, KLS09, ABL17, DKS18, DKTZ20c, DKK$^+$21a, DKTZ22]. An arguably surprising conceptual implication of our results is that the complexity of reliable learning for Gaussian halfspaces is qualitatively closer to the semi-random case.

## 1.3 Preliminaries

We use lowercase boldface letters for vectors and capitalized boldface letters for matrices and tensors. We use $\langle \mathbf{x}, \mathbf{y} \rangle$ for the inner product between $\mathbf{x}, \mathbf{y} \in \mathbb{R}^d$. For $\mathbf{x}, \mathbf{s} \in \mathbb{R}^d$, we use $\text{proj}_{\mathbf{s}}(\mathbf{x}) \overset{\text{def}}{=} \frac{\langle \mathbf{x}, \mathbf{s} \rangle \mathbf{s}}{\|\mathbf{s}\|_2^2}$ for the projection of $\mathbf{x}$ on the $\mathbf{s}$ direction. Similarly, we use $\text{proj}_{\perp \mathbf{s}}(\mathbf{x}) = \mathbf{x} - \text{proj}_{\mathbf{s}}(\mathbf{x})$ for the projection of $\mathbf{x}$ on the orthogonal complement of $\mathbf{s}$. Additionally, let $\mathbf{x}^V \in \mathbb{R}^{\dim(V)}$ be the projection of $\mathbf{x}$ on the subspace $V$ and reparameterized on $\mathbb{R}^{\dim(V)}$. More precisely, let $\mathbf{B}_V \in \mathbb{R}^{d \times \dim(V)}$ be the matrix whose columns form an (arbitrary) orthonormal basis for the subspace $V$, and let $\mathbf{x}^V \overset{\text{def}}{=} (\mathbf{B}_V)^{\mathsf{T}} \mathbf{x}$. For $p \geq 1$ and $\mathbf{x} \in \mathbb{R}^d$, we use $\|\mathbf{x}\|_p \overset{\text{def}}{=} (\sum_{i=1}^n |\mathbf{x}_i|^p)^{1/p}$ to denote the $\ell_p$-norm of $\mathbf{x}$. For a matrix or tensor $\mathbf{T}$, we denote by $\|\mathbf{T}\|_F$ the Frobenius norm of $\mathbf{T}$.

We use $\mathcal{N}_d$ to denote the standard normal distribution $\mathcal{N}(\mathbf{0}, \mathbf{I})$, where $\mathbf{0}$ is the $d$-dimensional zero vector and $\mathbf{I}$ is the $d \times d$ identity matrix. We use $\mathbf{x} \sim D$ to denote a random variable with distribution $D$. For a random variable $\mathbf{x}$ (resp. a distribution $D$), we use $P_{\mathbf{x}}$ (resp. $P_D$) to denote the probability density function or probability mass function of the random variable $\mathbf{x}$ (resp. distribution $D$). We also use $\Phi : \mathbb{R} \mapsto [0, 1]$ to denote the cdf function of $\mathcal{N}_1$. We use $\mathbf{1}$ to denote the indicator function.

For a boolean function $h : \mathbb{R}^d \to \{\pm 1\}$ and a distribution $D$ supported on $\mathbb{R}^d \times \{\pm 1\}$, we use $R_+(h; D) \overset{\text{def}}{=} \mathbf{Pr}_{(\mathbf{x}, y) \sim D}[h(\mathbf{x}) = 1 \wedge y \neq 1]$ (resp. $R_-(h; D) \overset{\text{def}}{=} \mathbf{Pr}_{(\mathbf{x}, y) \sim D}[h(\mathbf{x}) = -1 \wedge y \neq -1]$) to denote the false positive (resp. false negative) 0-1 error.

## 2 Algorithm for Reliably Learning Gaussian Halfspaces

In this section, we describe and analyze our algorithm establishing Theorem 1.3. Due to space limitations, some proofs have been deferred to Appendix B. For convenience, we will assume that $\alpha$ (the bias of the optimal halfspace) is known to the algorithm and that the excess error satisfies $\epsilon \leq \alpha/3$. These assumptions are without loss of generality for the following reasons: First, one can efficiently reduce the unknown $\alpha$ case to the case that $\alpha$ is known, by guessing the value of $\alpha$. Second, if $\epsilon$ is large, there is a straightforward algorithm for the problem (simply output the best constant hypothesis).

**Notation:** We use the notation $\mathcal{H}_d^{\alpha}$ for the set of all LTFs on $\mathbb{R}^d$ whose bias is equal to $\alpha$. Given the above assumptions, it suffices for us to give a reliable learning algorithm for $\mathcal{H}_d^{\alpha}$ instead of $\mathcal{H}_d$.

The high-level idea of the algorithm is as follows. Without loss of generality, we assume that there exists an $\alpha$-biased halfspace that correctly classifies all the points with label $y = -1$—since otherwise the algorithm can just return the hypothesis $h(\mathbf{x}) \equiv -1$.

Let $f(\mathbf{x}) = \text{sign}(\langle \mathbf{w}^*, \mathbf{x} \rangle - t^*)$ be the optimal halfspace and let $\mathbf{w}$ be our current guess of $\mathbf{w}^*$. Assuming that $\mathbf{w}^*$ is not sufficiently close to $\mathbf{w}$ and the hypothesis that classifies all the points as negative is not optimal, we show that there exists a low-degree polynomial of the form $p(\langle \mathbf{w}, \mathbf{x} \rangle)$ with correlation at least $2^{-O(t^{*2})}\epsilon$ with the negative labels. By leveraging this structural result, we use a spectral algorithm to find a direction $\mathbf{v}$ that is non-trivially correlated with $\text{proj}_{\perp \mathbf{w}}(\mathbf{w}^*)$ with at least some constant probability.

Unfortunately, it is not easy to verify whether $\langle \mathbf{v}, \text{proj}_{\perp \mathbf{w}}(\mathbf{w}^*) \rangle$ is non-trivial. However, conditioned on the algorithm always getting a $\mathbf{v}$ that has good correlation, we show that it only takes at most $\log(1/\epsilon)^{O(t^{*2})}$ steps to get sufficiently close to $\mathbf{w}^*$. Therefore, repeating this process $2^{\log(1/\epsilon)^{O(t^{*2})}}$ many times will eventually find an accurate approximation to $\mathbf{w}^*$.

The structure of this section is as follows: In Section 2.1, we give our algorithm for finding a good direction. Section 2.2 describes and analyzes our random walk procedure.

## 2.1 Finding a Non-Trivial Direction

Here we present an algorithm that finds a direction that correlates non-trivially with the unknown optimal vector. We first show that there exists a zero-mean $O(\log(1/\alpha))$-degree polynomial that sign-matches the optimal hypothesis. Furthermore, using the fact that the optimal hypothesis always

correctly guesses the sign of the negative points, this gives us that the polynomial correlates with the negative (clean) points. Using this intuition, our algorithm estimates the first $O(\log(1/\alpha))$ moments of the distribution $D_{\mathbf{x}}$ conditioned on $y = -1$. This guarantees that at least one moment correlates with the optimal hypothesis, as a linear combination of the moments generates the sign-matching polynomial. Then, by taking a random vector that lies in the high-influence directions (which form a low-dimensional subspace), we guarantee that with constant probability, this vector correlates well with the unknown optimal vector. The main result of the section is the following.

**Proposition 2.1.** *Let $D$ be a joint distribution of $(\mathbf{x}, y)$ supported on $\mathbb{R}^d \times \{\pm 1\}$ with marginal $D_{\mathbf{x}} = \mathcal{N}_d$ and $\epsilon \in (0,1)$. Suppose $D$ satisfies the reliability condition with respect to $f(\mathbf{x}) = \mathrm{sign}(\langle \mathbf{w}^*, \mathbf{x} \rangle - t^*)$ with $t^* = O\left(\sqrt{\log(1/\epsilon)}\right)$ and $\mathbf{Pr}_{(\mathbf{x},y) \sim D}[y = -1] \geq \epsilon$. Then there is an algorithm that draws $N = d^{O(t^{*2})}/\epsilon^2$ samples, has $\mathrm{poly}(N)$ runtime, and with probability at least $\Omega(1)$ returns a unit vector $\mathbf{v}$ such that $\langle \mathbf{v}, \mathbf{w}^* \rangle \geq \max(\log(1/\epsilon)^{-O(t^{*2})}, \epsilon^{O(1)})$.*

We start by showing that for any distribution satisfying the reliability condition with respect to some $f(\mathbf{x}) = \mathrm{sign}(\langle \mathbf{w}^*, \mathbf{x} \rangle - t^*)$, there exists a degree-$O(t^{*2})$ zero-mean polynomial of the form $p(\langle \mathbf{w}^*, \mathbf{x} \rangle)$ that has correlation at least $2^{-O(t^{*2})} \mathbf{Pr}_{(\mathbf{x},y) \sim D}[y = -1]$ with the labels.

**Lemma 2.2** (Correlation with an Orthonormal Polynomial). *Let $D$ be a joint distribution of $(\mathbf{x}, y)$ supported on $\mathbb{R}^d \times \{\pm 1\}$ with marginal $D_{\mathbf{x}} = \mathcal{N}_d$. Suppose $D$ satisfies the reliability condition with respect to $f(\mathbf{x}) = \mathrm{sign}(\langle \mathbf{w}^*, \mathbf{x} \rangle - t^*)$. Then there exists a polynomial $p : \mathbb{R} \mapsto \mathbb{R}$ of degree at most $k = O(t^{*2} + 1)$ such that $\mathbf{E}_{z \sim \mathcal{N}_1}[p(z)] = 0$, $\mathbf{E}_{z \sim \mathcal{N}_1}[p^2(z)] = 1$ and $\mathbf{E}_{(\mathbf{x},y) \sim D}[y\, p(\langle \mathbf{w}^*, \mathbf{x} \rangle)] = 2^{-O(t^{*2})} \mathbf{Pr}_{(\mathbf{x},y) \sim D}[y = -1]$.*

*Proof Sketch of Lemma 2.2.* Note that since $p$ is a zero-mean polynomial with respect to $D_{\mathbf{x}}$, we have that $\mathbf{E}_{(\mathbf{x},y) \sim D}[y\, p(\langle \mathbf{w}^*, \mathbf{x} \rangle)] = -2\, \mathbf{E}_{(\mathbf{x},y) \sim D}[\mathbf{1}(y = -1)\, p(\langle \mathbf{w}^*, \mathbf{x} \rangle)]$ . Then, using the fact that the distribution $D$ satisfies the reliability condition, the statement boils down to showing that for any $\epsilon$-mass inside the interval $[t^*, \infty]$, the expectation of $p$ on this $\epsilon$ mass is at least $2^{-O(t^{*2})}$. To prove this statement, it suffices to construct a polynomial $p$ that is non-negative on $[t^*, \infty]$ and such that the $\epsilon/2$-tail of $p$ in that interval is at least $2^{-O(t^{*2})}$. To achieve this, we show that the sign-matching polynomial used in [DKK⁺22] meets our purpose. $\square$

Our algorithm uses the following normalized Hermite tensor.

**Definition 2.3** (Hermite Tensor). *For $k \in \mathbb{N}$ and $\mathbf{x} \in \mathbb{R}^d$, we define the $k$-th Hermite tensor as*

$$(\mathbf{H}_k(\mathbf{x}))_{i_1, i_2, \ldots, i_k} = \frac{1}{\sqrt{k!}} \sum_{\substack{\text{Partitions } P \text{ of } [k] \\ \text{into sets of size 1 and 2}}} \bigotimes_{\{a,b\} \in P} (-\mathbf{I}_{i_a, i_b}) \bigotimes_{\{c\} \in P} \mathbf{x}_{i_c} .$$

Given that there is such a polynomial, we show that if we take a flattened version of the Chow-parameter tensors (which turns them into matrices) and look at the space spanned by their top singular vectors, then a non-trivial fraction of $\mathbf{w}^*$ must lie inside this subspace. We prove the following lemma, which is similar to Lemma 5.10 in [DKK⁺22]. See Appendix B for the proof.

**Lemma 2.4.** *Let $D$ be the joint distribution of $(\mathbf{x}, y)$ supported on $\mathbb{R}^d \times \{\pm 1\}$ with marginal $D_{\mathbf{x}} = \mathcal{N}_d$. Let $p : \mathbb{R} \mapsto \mathbb{R}$ be a univariate, mean zero and unit variance polynomial of degree $k$ such that for some unit vector $\mathbf{v}^* \in \mathbb{R}^d$ it holds $\mathbf{E}_{(\mathbf{x},y) \sim D}[\mathbf{1}(y = -1) p(\langle \mathbf{v}^*, \mathbf{x} \rangle)] \geq \tau$ for some $\tau \in (0, 1]$. Let $\mathbf{T}'^m$ be an approximation of the order-$m$ Chow-parameter tensor $\mathbf{T}^m = \mathbf{E}_{(\mathbf{x},y) \sim D}[\mathbf{1}(y = -1)\mathbf{H}_m(\mathbf{x})]$ such that $\|\mathbf{T}'^m - \mathbf{T}^m\|_F \leq \tau/(4\sqrt{k})$. Denote by $V_m$ the subspace spanned by the left singular vectors of flattened $\mathbf{T}'^m$ whose singular values are greater than $\tau/(4\sqrt{k})$. Moreover, denote by $V$ the union of $V_1, \cdots, V_k$. Then, for $\gamma = \mathbf{Pr}_{(\mathbf{x},y) \sim D}[y = -1]$, we have that*

*1. $\dim(V) = O(\gamma^2 \log(1/\gamma)^k k^2/\tau^2 + 1)$, and*

*2. $\|\mathrm{proj}_V(\mathbf{v}^*)\|_2 = \Omega\left(\tau/\left(\sqrt{k}\gamma \log(1/\gamma)^{k/2}\right)\right)$.*

By Lemma 2.4, taking a random unit vector $\mathbf{v}$ in $V$ will give us $\langle \mathbf{v}, \mathbf{v}^* \rangle \geq \|\mathrm{proj}_V(\mathbf{v}^*)\|/\sqrt{\dim(V)}$. We are ready to prove Proposition 2.1. For the algorithm pseudocode, see Appendix B.

*Proof Sketch of Proposition 2.1 .* The idea is that the empirical estimate $\mathbf{T}'^m$ obtained using $d^{O(\max\{t^{*2},1\})}\log(1/\delta)/\epsilon^2$ many samples will satisfy $\|\mathbf{T}'^m - \mathbf{T}^m\|_F \leq \tau/(4\sqrt{k})$ with high probability. Then, by Lemma 2.2 and Lemma 2.4, if we take $\mathbf{v}$ to be a random unit vector in $V$, with constant probability we will have $\langle \mathbf{v}, \mathbf{w}^* \rangle = \log(1/\epsilon)^{-O(t^{*2})}$. For $\langle \mathbf{v}, \mathbf{w}^* \rangle = \epsilon^{O(1)}$, the proof relies on a different version of Lemma 2.4. $\qquad\square$

## 2.2 Random Walk to Update Current Guess

We now describe how we use Proposition 2.1 to construct an algorithm for our learning problem. Let $\mathbf{w}$ be the current guess for $\mathbf{w}^*$. For convenience, we assume that $\langle \mathbf{w}, \mathbf{w}^* \rangle = \Omega(1/\sqrt{d})$. Let $D'$ be the distribution of $\mathbf{x}^{\perp \mathbf{w}}$ conditioned on $\mathbf{x} \in B$, where $B = \{\mathbf{x} \in \mathbb{R}^d : \langle \mathbf{w}, \mathbf{x} \rangle - t^* \geq 0\}$. We next show that the $\mathbf{x}$-marginals of $D'$ are standard Gaussian and that $D'$ satisfies the reliability condition with respect to the halfspace $h(\mathbf{x}) = \text{sign}(\langle \mathbf{w}', \mathbf{x} \rangle - t')$ with $\mathbf{w}' = \mathbf{w}^{*\perp \mathbf{w}}$ and $|t'| \leq |t^*|$.

**Lemma 2.5.** *Let $D$ be the joint distribution of $(\mathbf{x}, y)$ supported on $\mathbb{R}^d \times \{\pm 1\}$ with marginal $D_{\mathbf{x}} = \mathcal{N}_d$ and $\epsilon \in (0,1)$. Suppose $D$ satisfies the reliability condition with respect to $f(\mathbf{x}) = \text{sign}(\langle \mathbf{w}^*, \mathbf{x} \rangle - t^*)$. Suppose that $h(\mathbf{x}) = \text{sign}(\langle \mathbf{w}, \mathbf{x} \rangle - t)$ with $\langle \mathbf{w}, \mathbf{w}^* \rangle > 0$ and $t - t^* \in [0, \epsilon/100]$ does not satisfy $R_h^+(D) \leq \epsilon/2$. Let $B = \{\mathbf{x} \in \mathbb{R}^d : \langle \mathbf{w}, \mathbf{x} \rangle - t \geq 0\}$ and $D'$ be the distribution of $(\mathbf{x}', y) = (\mathbf{x}^{\perp \mathbf{w}}, y)$ given $\mathbf{x} \in B$, then*

1. *$D'$ has marginal distribution $D_{\mathbf{x}'} = \mathcal{N}_{d-1}$,*

2. *$D'$ satisfies the reliability condition with respect to $h'(\mathbf{x}) = \text{sign}(\langle \mathbf{w}', \mathbf{x} \rangle - t')$, where $\mathbf{w}' = \mathbf{w}^{*\perp \mathbf{w}}/\|\mathbf{w}^{*\perp \mathbf{w}}\|_2$ and $|t'| \leq |t^*|$, and*

3. *$\mathbf{Pr}_{(\mathbf{x}',y)\sim D'}[y = -1] \geq \epsilon/2$.*

*Proof Sketch of Lemma 2.5.* Item 1 follows by definition. For Item 2, we consider two cases: $t^* > 0$; and $t^* \leq 0$. For the case $t^* \leq 0$, we prove that the distribution $D'$ satisfies the reliability condition with respect to $h'(\mathbf{x}) = \text{sign}(\langle \mathbf{w}', \mathbf{x} \rangle)$, where $\mathbf{w}' = \mathbf{w}^{*\perp \mathbf{w}}/\|\mathbf{w}^{*\perp \mathbf{w}}\|_2$. For the case $t^* > 0$, we prove that the distribution $D'$ satisfies the reliability condition with respect to $h'(\mathbf{x}) = \text{sign}(\langle \mathbf{w}', \mathbf{x} \rangle - t')$, where $\mathbf{w}' = \mathbf{w}^{*\perp \mathbf{w}}/\|\mathbf{w}^{*\perp \mathbf{w}}\|_2$ and $t' = t^*$. Then Item 3 follows from the fact that $h$ does not satisfy $R_h^+(D) \leq \epsilon/2$. $\qquad\square$

By Lemma 2.5, given any current guess $\mathbf{w}$ such that $h(\mathbf{x}) = \text{sign}(\langle \mathbf{w}, \mathbf{x} \rangle - t^*)$ does not satisfy $R_+(h; D) \leq \epsilon/2$, the corresponding distribution $D'$ satisfies the reliability condition with respect to $h'$ and $\mathbf{Pr}_{(\mathbf{x},y)\sim D'}[y = -1] \geq \epsilon/2$. Therefore, $D'$ satisfies the assumptions of Proposition 2.1. So, if we apply the algorithm in Proposition 2.1, we will with probability at least $\Omega(1)$ get a unit vector $\mathbf{v}$ such that $\langle \mathbf{v}, \mathbf{w} \rangle = 0$ and $\langle \mathbf{v}, \mathbf{w}^* \rangle = \log(1/\epsilon)^{-O(t^{*2})}$.

The following fact shows that by updating our current guess in the direction of $\mathbf{v}$ with appropriate step size, we can get an updated guess with increased correlation with $\mathbf{w}^*$.

**Fact 2.6** (Correlation Improvement, Lemma 5.13 in [DKK+22]). *Fix unit vectors $\mathbf{v}^*, \mathbf{v} \in \mathbb{R}^d$. Let $\mathbf{u} \in \mathbb{R}^d$ such that $\langle \mathbf{u}, \mathbf{v}^* \rangle \geq c$, $\langle \mathbf{u}, \mathbf{v} \rangle = 0$ and $\|\mathbf{u}\|_2 \leq 1$ with $c > 0$. Then, for $\mathbf{v}' = \frac{\mathbf{v}+\lambda\mathbf{u}}{\|\mathbf{v}+\lambda\mathbf{u}\|_2}$, with $\lambda = c/2$, we have that $\langle \mathbf{v}', \mathbf{v}^* \rangle \geq \langle \mathbf{v}, \mathbf{v}^* \rangle + \lambda^2/2$.*

Notice that because $\|\text{proj}_{\perp \mathbf{w}}(\mathbf{w}^*)\|_2$ is unknown, we cannot always choose the optimal step size $\lambda$. Instead, we will use the same $\lambda$ to do sufficiently many update steps such that after that many updates, we are certain that $\|\text{proj}_{\perp \mathbf{w}}(\mathbf{w}^*)\|_2 \leq 3\lambda$. We then take the new step size $\lambda_{\text{update}} = \lambda/2$ and repeat this process, until $\mathbf{w}$ and $\mathbf{w}^*$ are sufficiently close to each other.

We are now ready to describe our algorithm (Algorithm 1) and prove its correctness.

Notice that given that we know the bias $\alpha$, $t$ must be either $-\Phi^{-1}(\alpha)$ or $\Phi^{-1}(\alpha)$. For convenience, we assume that $t = \Phi^{-1}(\alpha)$. To account for the case that $t = -\Phi^{-1}(\alpha)$, we can simply run Algorithm 1 twice and pick the output halfspace with the smallest $t$ value (or even run a different efficient algorithm, since $t \leq 0$ as explained in Appendix B). For convenience, we also assume $\min\left(2^{\log(1/\epsilon)^{O(\log(1/\alpha))}}, 2^{\text{poly}(1/\epsilon)}\right) = 2^{\log(1/\epsilon)^{O(\log(1/\alpha))}}$. To account for the other case, we can simply initialize the step size $\zeta = \epsilon^c$ for a sufficiently large constant $c$ instead of $\zeta = \log(1/\epsilon)^{-ct^{*2}}$.

A more detailed version of Algorithm 1 is deferred to Appendix B.

---

**Input:** $\epsilon \in (0, 1)$, $\alpha \in (0, 1/2)$ and samples access to a joint distribution $D$ of $(\mathbf{x}, y)$ supported on $\mathbb{R}^d \times \{\pm 1\}$ with $\mathbf{x}$-marginal $D_{\mathbf{x}} = \mathcal{N}_d$.
**Output:** $h(\mathbf{x}) = \mathrm{sign}(\langle \mathbf{w}, \mathbf{x} \rangle - t)$ that is $\epsilon$-reliable with respect to the class $\mathcal{H}_d^\alpha$.

1. Check if $\mathbf{Pr}_{(\mathbf{x},y) \sim D}[y = -1] \leq \epsilon/2$ (with sufficiently small constant failure probability). If so, return the $+1$ constant hypothesis. Set the initial step size $\zeta = \log(1/\epsilon)^{-ct^{*2}}$, where $c$ is a sufficiently large universal constant and $t = \Phi^{-1}(\alpha)$.

2. Initialize $\mathbf{w}$ to be a random unit vector in $\mathbb{R}^d$. Let the update step size $\lambda = \zeta$ and repeat the following process until $\lambda \leq \epsilon/100$.

   (a) Use samples from $D$ to check if the hypothesis $h(\mathbf{x}) = \mathrm{sign}(\langle \mathbf{w}, \mathbf{x} \rangle - t)$ satisfies $R_+(h; D) \leq \epsilon/2$. If so, go to Step (3).

   (b) With $1/2$ probability, let $\mathbf{w} = -\mathbf{w}$. Let $B = \{\mathbf{x} \in \mathbb{R}^d : \langle \mathbf{w}, \mathbf{x} \rangle - t \geq 0\}$, and let $D'$ be the distribution of $(\mathbf{x}^{\perp \mathbf{w}}, y)$ for $(\mathbf{x}, y) \sim D$ given $\mathbf{x} \in B$. Use the algorithm of Proposition 2.1 on $D'$ to find a unit vector $\mathbf{v}$ such that $\langle \mathbf{v}, \mathbf{w} \rangle = 0$ and $\left\langle \mathbf{v}, \frac{\mathrm{proj}_{\perp \mathbf{w}}(\mathbf{w}^*)}{\|\mathrm{proj}_{\perp \mathbf{w}}(\mathbf{w}^*)\|_2} \right\rangle \geq \zeta$. Then update $\mathbf{w}$ as follows: $\mathbf{w}_{\mathrm{update}} = \frac{\mathbf{w} + \lambda \mathbf{v}}{\|\mathbf{w} + \lambda \mathbf{v}\|_2}$.

   (c) Repeat Steps (2a) and (2b) $c/\zeta^2$ times, where $c$ is a sufficiently large universal constant, with the same step size $\lambda$. After that, update the new step size as $\lambda_{\mathrm{update}} = \lambda/2$.

3. Check if $h(\mathbf{x}) = \mathrm{sign}(\langle \mathbf{w}, \mathbf{x} \rangle - t)$ satisfies $R_+(h; D) \leq \epsilon/2$. If so, return $h$ and terminate. Repeat Step (2) $2^{1/\zeta^c}$ many times where $c$ is a sufficiently large constant.

---

**Algorithm 1:** Reliably Learning General Halfspaces with Gaussian Marginals.

*Proof Sketch of the algorithmic part of Theorem 1.3.* Let $f(\mathbf{x}) = \mathrm{sign}(\langle \mathbf{w}^*, \mathbf{x} \rangle - t^*)$ be the optimal halfspace with $\alpha$ bias. We need to show that with high probability Algorithm 1 returns a hypothesis $h(\mathbf{x}) = \mathrm{sign}(\langle \mathbf{w}, \mathbf{x} \rangle - t)$ such that $R_+(h; D) \leq \epsilon$ and $R_-(h; D) \leq R_-(f; D) + \epsilon$.

To do so, it suffices to show that $R_+(h; D) \leq \epsilon$; given $R_+(h; D) \leq \epsilon$, $R_-(h; D) \leq R_-(f; D) + \epsilon$ follows from our choice of $t$. For convenience, we can assume $h$ never satisfies $R_+(h; D) \leq \epsilon/2$ in Step (2a) (otherwise, we are done). We can also assume that the subroutine in Proposition 2.1 always succeeds since the algorithm repeats Step (2) sufficiently many times. Given the above conditions, using Fact 2.6, one can show that each time after $c/\zeta^2$ many updates in Step (2b), we must have $\|\mathrm{proj}_{\perp \mathbf{w}} \mathbf{w}^*\|_2 \leq 3\lambda$. Therefore, when we have $\lambda \leq \epsilon/100$, then $\|\mathrm{proj}_{\perp \mathbf{w}} \mathbf{w}^*\|_2 \leq 3\epsilon/100$, which implies $R_+(h; D) \leq \epsilon/2$. □

## 3 Nearly Matching SQ Lower Bound

In this section, we establish the SQ hardness result of Theorem 1.3. Due to space limitations, some proofs have been deferred to Appendix C.

**Proof Overview** To establish our SQ lower bound for reliable learning, we first prove an SQ lower bound for a natural decision version of reliably learning $\alpha$-biased LTFs. We define the following decision problem over distributions.

**Definition 3.1** (Decision Problem over Distributions). *Let $D$ be a fixed distribution and $\mathcal{D}$ be a distribution family. We denote by $\mathcal{B}(\mathcal{D}, D)$ the decision problem in which the input distribution $D'$ is promised to satisfy either (a) $D' = D$ or (b) $D' \in \mathcal{D}$, and the goal is to distinguish the two cases with high probability.*

We show that given SQ access to a joint distribution $D$ of $(\mathbf{x}, y)$ supported on $\mathbb{R}^d \times \{\pm 1\}$ with marginal $D_{\mathbf{x}} = \mathcal{N}(\mathbf{0}, \mathbf{I})$, it is hard to solve the problem $\mathcal{B}(\mathcal{D}, D)$ with the following distributions.

(a) Null hypothesis: $D$ is the distribution so that $y = 1$ with probability $1/2$ independent of $\mathbf{x}$.

(b) Alternative hypothesis: $D \in \mathcal{D}$, where $\mathcal{D}$ is a family of distributions such that for any distribution $D \in \mathcal{D}$, there exists an $\alpha$-biased LTF $f$ such that $R_+(f; D) = 0$.

In order to construct such a family of distributions $\mathcal{D}$, we start by constructing a joint distribution $D'$ of $(z, y)$ over $\mathbb{R} \times \{\pm 1\}$ such that the marginal distribution of $z$ is $\mathcal{N}_1$ and the conditional distributions $z \mid y = 1$ and $z \mid y = -1$ both match many moments with the standard Gaussian $\mathcal{N}_1$. Moreover, there is $\alpha$ probability mass on the positive side of the marginal distribution of $z$ that is purely associated with $y = 1$ (i.e., $\mathbf{E}_{(z,y) \sim D}[y \mid z \geq c] = 1$ where $c = \Phi^{-1}(1 - \alpha)$). We then embed this distribution $D$ along a hidden direction inside the joint distribution of $(\mathbf{x}, y)$ on $\mathbb{R}^d \times \{\pm 1\}$ to construct a family of hard-to-distinguish distributions using the "hidden-direction" framework developed in [DKS17] and enhanced in [DKPZ21, DKRS23].

We can now proceed with the details of the proof. We start by defining the pairwise correlation between distributions.

**Definition 3.2** (Pairwise Correlation). *The pairwise correlation of two distributions with pdfs $D_1, D_2 : \mathbb{R}^d \mapsto \mathbb{R}_+$ with respect to a distribution with density $D : \mathbb{R}^d \mapsto \mathbb{R}_+$, where the support of $D$ contains the support of $D_1$ and $D_2$, is defined as $\chi_D(D_1, D_2) \stackrel{\text{def}}{=} \int_{\mathbb{R}^d} D_1(\mathbf{x}) D_2(\mathbf{x}) / D(\mathbf{x}) d\mathbf{x} - 1$. Furthermore, the $\chi$-squared divergence of $D_1$ to $D$ is defined as $\chi^2(D_1, D) \stackrel{\text{def}}{=} \chi_D(D_1, D_1)$.*

In particular, the framework in [DKS17] allows us to construct a family of $2^{d^{\Omega(1)}}$ distributions on $\mathbb{R}^d \times \{\pm 1\}$ whose pairwise correlation is $d^{-\Omega(n)}$ where $n$ is the number of matching moments $D'$ has with the standard Gaussian. Then, using standard SQ dimension techniques, this gives an SQ lower bound for the distinguishing problem. After that, we reduce the distinguishing problem to the problem of reliably learning $\alpha$-biased LTFs under Gaussian marginals with additive error $\epsilon < \alpha/3$.

In order to construct such a distribution $D'$ of $(z, y)$ supported on $\mathbb{R} \times \{\pm 1\}$, we reparameterize $\mathbf{E}_{(z,y) \sim D'}[y|z]$ as $g(z)$. For a function $g : \mathbb{R} \to \mathbb{R}$, we use $\|g\|_p = \mathbf{E}_{t \sim \mathcal{N}_1}[|g(t)|^p]^{1/p}$ for its $L_p$ norm. We let $L^1(R)$ denote the set of all functions $g : \mathbb{R} \to \mathbb{R}$ that have finite $L_1$-norm.

We use linear programming over one-dimensional functions in $L^1(\mathbb{R})$ space to establish the existence of such a function. Specifically, we show the following:

**Lemma 3.3.** *For any sufficiently large $n \in \mathbb{N}$, there exists a function $g : \mathbb{R} \mapsto [-1, +1]$ such that $g$ satisfies the following properties:*

*(i) $g(z) = 1$ for all $z \geq c$, where $c = \Phi^{-1}(1 - 3^{-2n}/4)$, and*

*(ii) $\mathbf{E}_{t \sim \mathcal{N}_1}[g(z)z^k] = 0$ for all $k \in [n]$.*

*Proof Sketch of Lemma 3.3.* We let $P_n$ denote the set of all polynomials $p : \mathbb{R} \to \mathbb{R}$ of degree at most $n$ and let $L^1_+(\mathbb{R})$ denote the set of all nonnegative functions in $L^1(\mathbb{R})$. Then, using linear programming, we will get the following primal:

$$
\begin{aligned}
&\text{find} && g \in L^1(\mathbb{R}) \\
&\text{such that} && \mathop{\mathbf{E}}_{z \sim \mathcal{N}_1}[p(z)g(z)] = 0 , && \forall p \in P_n \\
& && \mathop{\mathbf{E}}_{z \sim \mathcal{N}_1}[g(z)h(z)\mathbb{1}\{t \geq c\}] \geq \|h(z)\mathbb{1}\{z \geq c\}\|_1 , && \forall h \in L^1_+(\mathbb{R}) \\
& && \mathop{\mathbf{E}}_{z \sim \mathcal{N}_1}[g(z)H(z)] \leq \|H\|_1 , && \forall H \in L^1(\mathbb{R})
\end{aligned}
$$

Then, using (infinite-dimensional) LP duality, we get that the above primal is feasible if and only if there is no polynomial of degree $n$ such that $\mathbf{E}_{t \sim \mathcal{N}_1}[|p(t)|\mathbb{1}(t \leq c)] < \mathbf{E}_{t \sim \mathcal{N}_1}[p(t)\mathbb{1}(t \geq c)]$.

Using Gaussian hypercontractivity (see [Bog98, Nel73]), one can show that for every polynomial $p$ and $c \in \mathbb{R}$, it holds

$$
\mathop{\mathbf{E}}_{z \sim \mathcal{N}_1}[|p(z)|] < 2 \cdot 3^n \mathop{\mathbf{E}}_{z \sim \mathcal{N}_1}[|p(z)|] \left( \mathop{\mathbf{Pr}}_{z \sim \mathcal{N}_1}[z \geq c] \right)^{1/2} .
$$

From our choice of the parameter $c$, we have that $\mathbf{Pr}_{z \sim \mathcal{N}_1}[z \geq c] \leq 3^{-2n}/4$; thus, $\mathbf{E}_{z \sim \mathcal{N}_1}[|p(z)|] < \mathbf{E}_{z \sim \mathcal{N}_1}[|p(z)|]$, which is a contradiction. Therefore, such a polynomial $p$ cannot exist. $\qquad \square$

We have proven the existence of the function $g$ in Lemma 3.3. Now, we construct a joint distribution of $(z, y)$ on $\mathbb{R} \times \{\pm 1\}$ such that $\mathbf{E}[y|z]$ is exactly $g$, as we discussed in the proof outline. For a joint distribution $D$ of $(x, y)$ supported on $X \times \{\pm 1\}$, we will use $D_+$ to denote the conditional distribution of $x$ given $y = 1$; and $D_-$ for the distribution of $x$ given $y = -1$.

**Lemma 3.4.** *For any sufficiently large $n \in \mathbb{N}$, there exists a distribution $D$ on $\mathbb{R} \times \{\pm 1\}$ such that for $(z, y) \sim D$:*

(i) *the marginal distribution $D_z = \mathcal{N}_1$;*

(ii) $\mathbf{E}_{(z,y) \sim D}[y|z = z'] = 1$ *for all $z' \geq \Phi^{-1}(1 - 3^{-2n}/4)$;*

(iii) $\mathbf{E}_{(z,y) \sim D}[y] = 0$ *and* $\mathbf{E}_{(z,y) \sim D}[z^k] = \mathbf{E}_{(z,y) \sim D}[z^k \mid y = 1] = \mathbf{E}_{(z,y) \sim D}[z^k \mid y = -1]$ *for all $k \in [n]$;*

(iv) $\chi^2(D_+, \mathcal{N}_1), \chi^2(D_-, \mathcal{N}_1) = O(1)$.

*Proof Sketch for Lemma 3.4.* The properties here directly follow from the properties of $g$. $\qquad \square$

Using the framework introduced in [DKS17, DKPZ21], we can construct a set of alternative hypothesis distributions $\mathcal{D} = \{D_\mathbf{v} : \mathbf{v} \in V\}$ on $\mathbb{R}^d \times \{\pm 1\}$, where $V$ is a set of exponentially many pairwise nearly-orthogonal vectors and the marginal distribution of each $D_\mathbf{v}$ on direction $\mathbf{v}$ is the distribution $D$ in Lemma 3.4. This effectively embeds the distribution $D$ in a hidden direction $\mathbf{v}$. The size of the family $\mathcal{D}$ is exponential in $d$, and the distributions in it have small pairwise correlations. The details of $\mathcal{D}$ are deferred to Appendix C. Now, we are ready to give a proof sketch for the SQ hardness part of our main theorem Theorem 1.3.

*Proof Sketch of the SQ hardness part of Theorem 1.3.* Let $\mathcal{D}$ be the set of distributions discussed above. We also let $D_{\text{null}}$ be the joint distribution of $(\mathbf{x}, y)$ such that $\mathbf{x} \sim \mathcal{N}_d$ and $y \sim \text{Bern}(1/2)$ independent of $\mathbf{x}$. Then, using standard SQ dimension techniques, one can show that any SQ algorithm that solves $\mathcal{B}(\mathcal{D}, D_{\text{null}})$ requires either queries of tolerance at most $d^{-\Omega(\log \frac{1}{\alpha})}$ or makes at least $2^{d^{\Omega(1)}}$ queries. By reducing the decision problem $\mathcal{B}(\mathcal{D}, D_{\text{null}})$ to reliably learning $\alpha$-biased LTFs with $\epsilon < \alpha/3$ accuracy, we get the lower bound part of the statement in Theorem 1.3. $\qquad \square$

# 4 Conclusions and Open Problems

In this paper, we study the problem of learning halfspaces under Gaussian marginals in the reliable learning model. Our main contribution is the design of the first efficient learner for this task whose complexity beats the complexity of agnostically learning this concept class. Moreover, we provide rigorous evidence, via an SQ lower bound, that no fully-polynomial time algorithm exists for general halfspaces. The obvious open question is whether the dependence on $\epsilon$ in the complexity of our algorithm can be improved. Specifically, is it possible to design a reliable learner with complexity $d^{O(\log(1/\alpha))}\text{poly}(1/\epsilon)$? Is it possible to obtain similarly efficient reliable learners under more general marginal distributions (e.g., strongly log-concave or discrete distributions)? More broadly, it would be interesting to characterize the computational separation between (distribution-specific) reliable and agnostic learning for other natural concept classes.

# 5    Acknowledgements

ID was supported in part by NSF Medium Award CCF-2107079 and an H.I. Romnes Faculty Fellowship. LR was supported in part by NSF Medium Award CCF-2107079. NZ was supported in part by NSF Medium Award CCF-2107079.

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

# Appendix

**Organization**   The supplementary material is structured as follows: Appendix A includes additional preliminaries. Appendix B includes the omitted proofs from Section 2. Finally, Appendix C includes the omitted proofs from Section 3.

## A   Additional Preliminaries

We use $\mathbb{S}^{d-1}$ to denote the $d$-dimensional unit sphere, i.e., $\mathbb{S}^{d-1} \overset{\text{def}}{=} \{\mathbf{x} \in \mathbb{R}^d : \|\mathbf{x}\|_2 = 1\}$.

**Basics of the SQ Model**   In order to give an SQ lower bound for reliable learning, we first present the basics of the SQ model. We define the pairwise correlation, which is used for the SQ lower bound.

**Definition A.1** (Pairwise Correlation). *The pairwise correlation of two distributions with probability density function $D_1, D_2 : \mathbb{R}^d \mapsto \mathbb{R}_+$ with respect to a distribution with density $D : \mathbb{R}^d \mapsto \mathbb{R}_+$, where the support of $D$ contains the support of $D_1$ and $D_2$, is defined as $\chi_D(D_1, D_2) \overset{\text{def}}{=} \int_{\mathbb{R}^d} D_1(\mathbf{x}) D_2(\mathbf{x}) / D(\mathbf{x}) d\mathbf{x} - 1$. Furthermore, the $\chi$-squared divergence of $D_1$ to $D$ is defined as $\chi^2(D_1, D) \overset{\text{def}}{=} \chi_D(D_1, D_1)$.*

We require the following definitions and lemmata from [FGR$^+$17] connecting pairwise correlation and SQ lower bounds.

**Definition A.2.** *We say that a set of $s$ distribution $\mathcal{D} = \{D_1, \cdots, D_s\}$ over $\mathbb{R}^d$ is $(\gamma, \beta)$-correlated relative to a distribution $D$ if $\chi_D(D_i, D_j) \leq \gamma$ for all $i \neq j$, and $\chi_D(D_i, D_j) \leq \beta$ for $i = j$.*

**Definition A.3** (Statistical Query Dimension). *For $\beta, \gamma > 0$, a decision problem $\mathcal{B}(\mathcal{D}, D)$, where $D$ is a fixed distribution and $\mathcal{D}$ is a family of distribution, let $s$ be the maximum integer such that there exists a finite set of distributions $\mathcal{D}_D \subseteq \mathcal{D}$ such that $\mathcal{D}_D$ is $(\gamma, \beta)$-correlated relative to $D$ and $|\mathcal{D}_D| \geq s$. The Statistical Query dimension with pairwise correlations $(\gamma, \beta)$ of $\mathcal{B}$ is defined to be $s$, and denoted by $s = \mathrm{SD}(\mathcal{B}, \gamma, \beta)$.*

**Lemma A.4.** *Let $\mathcal{B}(\mathcal{D}, D)$ be a decision problem, where $D$ is the reference distribution and $\mathcal{D}$ is a class of distribution. For $\gamma, \beta > 0$, let $s = \mathrm{SD}(\mathcal{B}, \gamma, \beta)$. For any $\gamma' > 0$, any SQ algorithm for $\mathcal{B}$ requires queries of tolerance at most $\sqrt{\gamma + \gamma'}$ or makes at least $s\gamma'/(\beta - \gamma)$ queries.*

**Basics of Hermite Polynomials**   We require the following definitions used in our work.

**Definition A.5** (Normalized Hermite Polynomial). *For $k \in \mathbb{N}$, we define the $k$-th probabilist's Hermite polynomials $He_k : \mathbb{R} \to \mathbb{R}$ as $He_k(t) = (-1)^k e^{t^2/2} \cdot \frac{\mathrm{d}^k}{\mathrm{d}t^k} e^{-t^2/2}$. We define the $k$-th normalized Hermite polynomial $h_k : \mathbb{R} \to \mathbb{R}$ as $h_k(t) = He_k(t)/\sqrt{k!}$.*

Furthermore, we use multivariate Hermite polynomials in the form of Hermite tensors (as the entries in the Hermite tensors are re-scaled multivariate Hermite polynomials). We define the *Hermite tensor* as follows.

**Definition A.6** (Hermite Tensor). *For $k \in \mathbb{N}$ and $\mathbf{x} \in \mathbb{R}^d$, we define the $k$-th Hermite tensor as*

$$(\mathbf{H}_k(\mathbf{x}))_{i_1, i_2, \ldots, i_k} = \frac{1}{\sqrt{k!}} \sum_{\substack{\text{Partitions } P \text{ of } [k] \\ \text{into sets of size 1 and 2}}} \bigotimes_{\{a,b\} \in P} (-\mathbf{I}_{i_a, i_b}) \bigotimes_{\{c\} \in P} \mathbf{x}_{i_c} .$$

We also point out that fully reliable learning (see [KKM12] for the definition) reduces to one-sided reliable learning (Definition 1.1). The proof is implicit in the proof of Theorem 3 in [KKM12] and we include it here for completeness.

**Fact A.7.** *For a joint distribution $D$ of $(\mathbf{x}, y)$ supported on $X \times \{\pm 1\}$, let $h_+ : X \to \{\pm 1\}$ (resp. $h_- : X \to \{\pm 1\}$) be a hypothesis that is $\epsilon/4$-reliable with respect to the concept class $C$ on distribution $D$ for positive labels (resp. for negative labels). Let hypothesis $h$ be defined as $h(\mathbf{x}) = 1$ if $h_+(\mathbf{x}) = h_-(\mathbf{x}) = 1$, $h(\mathbf{x}) = -1$ if $h_+(\mathbf{x}) = h_-(\mathbf{x}) = -1$ and $h(\mathbf{x}) = ?$ otherwise. Then $h$ is $\epsilon$ close to the best fully reliable sandwich hypothesis for $C$ on distribution $D$.*

*Proof Sketch.* We first show that $R_+(h; D), R_-(h; D) \leq \epsilon$. Notice that

$$R_+(h; D) = \Pr_{(\mathbf{x},y)\sim D}[h(\mathbf{x}) = 1 \wedge y = -1] \leq \Pr_{(\mathbf{x},y)\sim D}[h_+(\mathbf{x}) = 1 \wedge y = -1] \leq \epsilon \,.$$

The same holds for negative labels.

To show that $\Pr_{(\mathbf{x},y)\sim D}[h(\mathbf{x}) =?]$ is $\epsilon$-suboptimal, assume (for the purpose of contradiction) that there is an $h'$ such that $R_+(h'; D), R_-(h'; D) \leq \epsilon$ and $\Pr_{(\mathbf{x},y)\sim D}[h'(\mathbf{x}) =?] < \Pr_{(\mathbf{x},y)\sim D}[h(\mathbf{x}) =?] - \epsilon$. Notice that

$$
\begin{aligned}
&\Pr_{(\mathbf{x},y)\sim D}[h(\mathbf{x}) =?] \\
&= \Pr_{(\mathbf{x},y)\sim D}[h_+(\mathbf{x}) = 1 \wedge h_-(\mathbf{x}) = -1] + \Pr_{(\mathbf{x},y)\sim D}[h_+(\mathbf{x}) = -1 \wedge h_-(\mathbf{x}) = 1] \\
&\leq 1 - \Pr_{(\mathbf{x},y)\sim D}[h_+(\mathbf{x}) = 1] - \Pr_{(\mathbf{x},y)\sim D}[h_-(\mathbf{x}) = -1] + \Pr_{(\mathbf{x},y)\sim D}[h_+(\mathbf{x}) = 1 \wedge h_-(\mathbf{x}) = -1] \\
&\leq 1 - \Pr_{(\mathbf{x},y)\sim D}[h_+(\mathbf{x}) = 1] - \Pr_{(\mathbf{x},y)\sim D}[h_-(\mathbf{x}) = -1] + \epsilon/2 \,.
\end{aligned}
$$

Thus,

$$\Pr_{(\mathbf{x},y)\sim D}[h_+(\mathbf{x}) = 1] + \Pr_{(\mathbf{x},y)\sim D}[h_-(\mathbf{x}) = -1] \leq 1 - \Pr_{(\mathbf{x},y)\sim D}[h(\mathbf{x}) =?] + \epsilon/2 \,. \tag{1}$$

Then, define $h'_+ : X \to \{\pm 1\}$ as $h'_+(\mathbf{x}) = 1$ if $h'(\mathbf{x}) = 1$ and $h'_+(\mathbf{x}) = -1$ otherwise ($h'_-$ is defined similarly). We get $\Pr_{(\mathbf{x},y)\sim D}[h'_+(\mathbf{x}) = 1] + \Pr_{(\mathbf{x},y)\sim D}[h'_-(\mathbf{x}) = -1] = 1 - \Pr_{(\mathbf{x},y)\sim D}[h'(\mathbf{x}) =?]$. Using Equation (1) and $\Pr_{(\mathbf{x},y)\sim D}[h'(\mathbf{x}) =?] < \Pr_{(\mathbf{x},y)\sim D}[h(\mathbf{x}) =?] - \epsilon$, we get

$$\Pr_{(\mathbf{x},y)\sim D}[h_+(\mathbf{x}) = 1] - \Pr_{(\mathbf{x},y)\sim D}[h_-(\mathbf{x}) = -1] < \Pr_{(\mathbf{x},y)\sim D}[h'_+(\mathbf{x}) = 1] - \Pr_{(\mathbf{x},y)\sim D}[h'_-(\mathbf{x}) = -1] - \epsilon/2 \,.$$

Therefore, either $\Pr_{(\mathbf{x},y)\sim D}[h_+(\mathbf{x}) = 1] < \Pr_{(\mathbf{x},y)\sim D}[h'_+(\mathbf{x}) = 1] - \epsilon/4$ or

$$\Pr_{(\mathbf{x},y)\sim D}[h_-(\mathbf{x}) = -1] < \Pr_{(\mathbf{x},y)\sim D}[h'_-(\mathbf{x}) = -1] - \epsilon/4 \,.$$

Thus, either $h_+$ or $h_-$ is not $\epsilon/4$-suboptimal. This gives a contradiction. $\qquad\square$

# B  Omitted Proofs from Section 2

We define the bias of a function $h : \mathbb{R}^d \to \{\pm 1\}$ as $\min(\Pr_{\mathbf{x}\sim\mathcal{N}_d}[h(\mathbf{x}) = 1], \Pr_{\mathbf{x}\sim\mathcal{N}_d}[h(\mathbf{x}) = -1])$, and define $\mathcal{H}_d^\alpha$ to be the set of all LTFs whose bias is $\alpha$. Let $f = \operatorname{argmin}_{c\in\mathcal{H}_d^\alpha \,\wedge\, R_+(f;D)=0} R_-(f; D)$ be the optimal reliable hypothesis and $\alpha$ be the bias of $f$.

## B.1  Reliably Learning Halfspaces with Gaussian Marginals when $\epsilon \geq 2\alpha$

We show that for the problem in Theorem 1.3, in the case of $\epsilon \geq 2\alpha$, the algorithm can just return one of the constant hypotheses. Let $h_+$ (resp. $h_-$) be the $+1$ (resp. $-1$) constant hypothesis.

**Fact B.1.** *For the problem defined in Theorem 1.3, the case where $\epsilon \geq 2\alpha$ can be efficiently solved by returning $h_+$ if $R_+(h_+; D) \leq 3\epsilon/4$ and returning $h_-$ otherwise.*

*Proof.* When the algorithm returns $h_+$, it is easy to see that $h_+$ satisfies the reliable learning requirement in Definition 1.1. So we only consider the case that the algorithm returns $h_-$.

Given the algorithm returns $h_-$, either there is an $\alpha$-biased LTF $f$ such that $R_+(f; D) = 0$ or none of the $\alpha$-biased LTFs $f$ satisfies $R_+(f; D) = 0$. For the case that none of the $\alpha$-biased LTFs $f$ satisfies $R_+(f; D) = 0$, it is easy to see $h_-$ is a valid answer from the definition of Definition 1.1. For the case that there is an $\alpha$-biased LTF $f$ such that $R_+(f, D) = 0$, since the algorithm did not return $h_+$, it must be the case that $\Pr_{(\mathbf{x},y)\sim D}[y = -1] > 3\epsilon/4 > \alpha$. Therefore, we have $\Pr_{\mathbf{x}\sim\mathcal{N}_d}[f(\mathbf{x}) = -1] = 1 - \alpha$. Thus, we get $R_+(h_-, D) = 0$ and $R_-(h_-, D) \leq R_-(f, D) + \alpha$. This shows $h_-$ is $\epsilon$ reliable with respect to all $\alpha$ biased LTFs. This completes the proof. $\qquad\square$

## B.2 Reliably Learning Halfspaces with Gaussian Marginals: Case of Unknown $\alpha$

Here we establish our main result:

**Theorem B.2** (Main Algorithmic Result). *Let $\epsilon \in (0, 1/2)$ and let $D$ be a joint distribution of $(\mathbf{x}, y)$ supported on $\mathbb{R}^d \times \{\pm 1\}$ with marginal $D_\mathbf{x} = \mathcal{N}_d$. Let $f = \operatorname{argmin}_{f \in \mathcal{H}_d \,\wedge\, R_+(f;D)=0} R_-(f; D)$ be the optimal halfspace and $\alpha$ be its bias which is unknown. Then there is an algorithm that uses $N = d^{O(\log(\min\{1/\alpha, 1/\epsilon\}))} \min\left(2^{\log(1/\epsilon)^{O(\log(1/\alpha))}}, 2^{\operatorname{poly}(1/\epsilon)}\right)$ many samples from $D$, runs in $\operatorname{poly}(N, d, 1/\epsilon)$ time and with high probability returns a hypothesis $h(\mathbf{x}) = \operatorname{sign}(\langle \mathbf{w}, \mathbf{x} \rangle - t)$ that is $\epsilon$-reliable with respect to the class of LTFs.*

To prove Theorem B.2, we need to use the algorithm in the following lemma as a black box. The lemma shows that if the optimal halfspace $f$ satisfies $\mathbf{Pr}_{\mathbf{x} \sim \mathcal{N}_d}[f(\mathbf{x}) = -1] \leq 1/2$, then the problem can be solved efficiently.

**Lemma B.3.** *Let $D$ be the joint distribution of $(\mathbf{x}, y)$ supported on $\mathbb{R}^d \times \{\pm 1\}$ with marginal $D_\mathbf{x} = \mathcal{N}_d$ and $\epsilon \in (0, 1)$. Suppose the optimal halfspace $f = \operatorname{argmin}_{f \in \mathcal{H}_d \,\wedge\, R_+(f;D)=0} R_-(f; D)$ satisfies $\mathbf{Pr}_{\mathbf{x} \sim \mathcal{N}_d}[f(\mathbf{x}) = -1] \leq 1/2$. Then there is an algorithm that reliably learns LTFs on $D$ using $N = O(d/\epsilon^2)$ many samples and $\operatorname{poly}(N, d)$ running time.*

*Proof.* The algorithm is the following:

1. First check if $\mathbf{Pr}_{(\mathbf{x}, y) \sim D)}[y = -1] \leq \epsilon$ with sufficiently small constant failure probability. If so, return the $+1$ constant hypothesis.

2. Otherwise, draw $m = (d/\epsilon)^c$ many samples $S = \{(\mathbf{x}_1, y_1), \cdots, (\mathbf{x}_m, y_m)\}$ conditioned on $y = -1$. By Step 1, the sampling efficiency here is $\Omega(\epsilon)$ with high probability.

3. Solve the following semidefinite program for $\mathbf{w}'$ and $t'$:

$$
\begin{aligned}
\text{minimize} \qquad & t' \\
\text{such that} \quad \langle \mathbf{w}', \mathbf{x} \rangle - t' &\leq 0, \quad \forall (\mathbf{x}, y) \in S \\
\|\mathbf{w}'\|_2 &\leq 1
\end{aligned}
$$

Return the hypothesis $h(\mathbf{x}) = \operatorname{sign}(\langle \mathbf{w}, \mathbf{x} \rangle - t)$, where $\mathbf{w} = \mathbf{w}'/\|\mathbf{w}'\|_2$ and $t = t'/\|\mathbf{w}'\|_2$.

Let $f(\mathbf{x}) = \operatorname{sign}(\langle \mathbf{w}^*, \mathbf{x} \rangle - t^*)$ be the optimal hypothesis. We prove that $h(\mathbf{x}) = \operatorname{sign}(\langle \mathbf{w}, \mathbf{x} \rangle - t)$ is such that $R_+(h; D) \leq \epsilon$ and $R_-(h; D) \leq R_-(f; D) + \epsilon$. Since $h$ is consistent with all the negative samples, we have $R_+(h; D) \leq \epsilon$. From the assumption that $\mathbf{Pr}_{\mathbf{x} \sim \mathcal{N}_d}[f(\mathbf{x}) = -1] \leq 1/2$, we have $t^* \leq 0$. Notice that $\mathbf{w}^*$ and $t^*$ is a feasible solution to the above program; therefore, $t' \leq t^* \leq 0$ and we must have $t = t'/\|\mathbf{w}'\|_2 \leq t' \leq t^*$, which implies that $\mathbf{Pr}_{(\mathbf{x}, y) \sim D}[h(\mathbf{x}) = 1] \geq \mathbf{Pr}_{(\mathbf{x}, y) \sim D}[f(\mathbf{x}) = 1]$. Therefore,

$$
\begin{aligned}
R_-(h; D) &= \Pr_{(\mathbf{x}, y) \sim D}[y = 1] - \Pr_{(\mathbf{x}, y) \sim D}[y = 1 \wedge h(\mathbf{x}) = 1] \\
&= \Pr_{(\mathbf{x}, y) \sim D}[y = 1] - \Pr_{(\mathbf{x}, y) \sim D}[h(\mathbf{x}) = 1] + R_+(h; D) \\
&\leq \Pr_{(\mathbf{x}, y) \sim D}[y = 1] - \Pr_{(\mathbf{x}, y) \sim D}[f(\mathbf{x}) = 1] + \epsilon \\
&= R_-(f; D) + \epsilon .
\end{aligned}
$$

This completes the proof. $\qquad\square$

We now prove Theorem B.2.

*Proof of Theorem B.2.* The algorithm is the following:

1. Run the algorithm in Lemma B.3 with $\epsilon' = \epsilon/2$. If the output hypothesis $h$ satisfies $R_+(h; D) \leq \epsilon$ with a sufficiently small constant failure probability, then output $h$ and terminate.

2. Set $\alpha = 1/2 - \epsilon/100$ and run the algorithm in Theorem B.4 with $\epsilon' = \epsilon/2$. If the output hypothesis $h$ satisfies $R_+(h; D) \leq \epsilon$ with a sufficiently small constant failure probability, then output $h$ and terminate. Otherwise, update $\alpha$ as $\alpha - \epsilon/100$. Repeat Step 2 until the algorithm terminates.

Let $f = \mathrm{argmin}_{f \in \mathcal{H}_d \, \wedge \, R_+(f; D) = 0} R_-(f; D)$ be the optimal halfspace and $\alpha_f$ be its bias which is unknown. Suppose the algorithm terminates. Let $\alpha$ be the bias of the output hypothesis $h$. It is easy to see that $R_+(h; D) \leq \epsilon$ with high probability; therefore, it only remains to show $R_-(h; D) \leq R_-(f; D) + \epsilon$. Notice that if $\mathbf{Pr}_{\mathbf{x} \sim \mathcal{N}_d}[f(\mathbf{x}) = -1] \leq 1/2$, the algorithm will terminate in Step 1. Therefore, without loss of generality, we assume $\mathbf{Pr}_{\mathbf{x} \sim \mathcal{N}_d}[f(\mathbf{x}) = -1] \geq 1/2$. Then, by Theorem B.4, it is easy to see that $\alpha \geq \alpha_f - \epsilon/100$ since given any $\alpha \leq \alpha_f$, Step 2 is guaranteed to terminate. Therefore, we have

$$\Pr_{\mathbf{x} \sim \mathcal{N}_d}[h(\mathbf{x}) = -1] \leq 1 - \alpha \leq 1 - \alpha_f + \epsilon/100 = \Pr_{\mathbf{x} \sim \mathcal{N}_d}[f(\mathbf{x}) = -1] + \epsilon/100 \,.$$

Thus,

$$\begin{aligned}
R_-(h; D) &= \Pr_{(\mathbf{x}, y) \sim D}[y = 1] - \Pr_{(\mathbf{x}, y) \sim D}[y = 1 \wedge h(\mathbf{x}) = 1] \\
&= \Pr_{(\mathbf{x}, y) \sim D}[y = 1] - \Pr_{(\mathbf{x}, y) \sim D}[h(\mathbf{x}) = 1] + R_+(h; D) \\
&\leq \Pr_{(\mathbf{x}, y) \sim D}[y = 1] - \Pr_{(\mathbf{x}, y) \sim D}[f(\mathbf{x}) = 1] + \epsilon \\
&= R_-(f; D) + \epsilon \,.
\end{aligned}$$

This completes the proof. $\qquad\qquad\square$

### B.3 Reliably Learning Halfspaces with Gaussian Marginals for the Case $\epsilon \leq \alpha/2$

For convenience, we assume that $\alpha$ is known. This can be assumed without loss of generality as we discussed in Theorem B.2.

We show the following:

**Theorem B.4.** *Let $\epsilon \in (0, 1/2), \alpha \in (0, 1/2]$ and let $D$ be a joint distribution of $(\mathbf{x}, y)$ supported on $\mathbb{R}^d \times \{\pm 1\}$ with marginal $D_{\mathbf{x}} = \mathcal{N}_d$. Assume that there is a half-space in $\mathcal{H}_d^\alpha$ that is reliable with respect to $D$. Then, there is an algorithm that uses $N = d^{O(\log(\min\{1/\alpha, 1/\epsilon\}))} \min\left(2^{\log(1/\epsilon)^{O(\log(1/\alpha))}}, 2^{\mathrm{poly}(1/\epsilon)}\right)$ many samples from $D$, runs in $\mathrm{poly}(N, d, 1/\epsilon)$ time and with high probability returns a hypothesis $h(\mathbf{x}) = \mathrm{sign}(\langle \mathbf{w}, \mathbf{x} \rangle - t)$ that is $\epsilon$-reliable with respect to the class of $\mathcal{H}_d^\alpha$.*

Below we provide the omitted proofs from Section 2.

### B.4 Proof of Lemma 2.2

Notice that for any polynomial $p$ such that $\mathbf{E}_{z \sim \mathcal{N}_1}[p(z)] = 0$, we have

$$\mathop{\mathbf{E}}_{(\mathbf{x}, y) \sim D}[y \, p(\langle \mathbf{w}^*, \mathbf{x} \rangle)] = -2 \mathop{\mathbf{E}}_{(\mathbf{x}, y) \sim D}[\mathbf{1}(y = -1) \, p(\langle \mathbf{w}^*, \mathbf{x} \rangle)] \,.$$

Therefore, it suffices for us to show that there is a zero mean and unit variance polynomial $p$ such that

$$\mathop{\mathbf{E}}_{(\mathbf{x}, y) \sim D}[\mathbf{1}(y = -1) \, p(\langle \mathbf{w}^*, \mathbf{x} \rangle)] = 2^{-O(t^{*2})} \Pr_{(\mathbf{x}, y) \sim D}[y = -1] \,.$$

Here we will consider the sign-matching polynomial from [DKK+22], which satisfies the following properties:

**Claim B.5.** *Let $b \in \mathbb{R}$ and $b \geq 4$. There exists a zero mean and unit variance polynomial $p : \mathbb{R} \to \mathbb{R}$ of degree $k = \Theta(b^2)$ such that*

1. *The sign of $p(z)$ matches $\mathrm{sign}(z - b)$, i.e. $\mathrm{sign}(p(z)) = \mathrm{sign}(z - b)$.*

2. *For any $z \leq b/2$, we have $|p(z)| = 2^{-O(k)}$.*

*Proof.* We consider the polynomial $p$ defined as:

$$\tilde{p}(z) = q(z) - \frac{q(b)}{r(b)} r(z) ,$$

where $q(z) = z^{3k}$, $r(z) = z^{2k} - (2k-1)!!$ and $k$ is a sufficiently large odd integer such that $2|b| \leq \sqrt{k} \leq 4\max(|b|, 1)$. We then take $p(z) = \tilde{p}(z)/\sqrt{\mathbf{E}_{u \sim \mathcal{N}_1}[\tilde{p}^2(u)]}$.

For convenience, we first note that from Stirling's approximation for $m \in \mathbb{Z}^+$,

$$(m/2)^m \leq (2m-1)!! \leq (2m)^m .$$

To prove that $\text{sign}(p(z)) = \text{sign}(z - b)$, we first show that $\frac{q(b)}{r(b)} \leq 0$. Since $q(b) \geq 0$, we just need to show $r(b) \leq 0$. Notice $r(b) = b^{2k} - (2k-1)!!$, since $2|b| \leq \sqrt{k}$ and $(2k-1)!! \geq (k/2)^k$, thus $r(b) \leq (\sqrt{k}/2)^{2k} - (k/2)^k \leq 0$. Therefore, considering $q(z) = z^{3k}$ and $r(z) = z^{2k} - (2k-1)!!$,

$$\tilde{p}(z) = q(z) - \frac{q(b)}{r(b)} r(z)$$

must be monotone increasing for $z \geq b$ and $p(z)$ must also be monotone increasing for $z \geq b$. Notice that $p(b) = 0$; therefore, $p(z) > 0$ for $z > b$. To show $p(z) < 0$ for $z < -b$, we prove it for the cases $z \in [-b, +b)$ and $z < -b$. For $z \in [-b, +b)$, notice that due to $\frac{q(b)}{r(b)} \leq 0$ and the definition of $q(z)$ and $r(z)$,

$$\tilde{p}(z) = q(z) - \frac{q(b)}{r(b)} r(z) \leq q(b) - \frac{q(b)}{r(b)} r(b) < p(b) = 0 .$$

Then, for $z \leq -b$, we show that $p(z)$ is monotone increasing. Notice that the derivative $\tilde{p}'(z) = kz^{2k-1}(3z^k - 2\frac{q(b)}{r(b)})$, for $z \leq -b < 0$, this is positive if and only if $3z^k - 2\frac{q(b)}{r(b)} < 0$. If we can show $b^k \geq -\frac{2}{3}q(b)/r(b)$, then it is immediate that for any $z \leq -b$, $3z^k - 2\frac{q(b)}{r(b)} < 0$. The condition $b^k \geq -\frac{2}{3}q(b)/r(b)$ can be further simplified to $(2k-1)!!/b^{2k} \geq 5/3$. Then considering $2b < \sqrt{k}$ and $(2k-1)!! \geq (k/2)^k$, we have

$$(2k-1)!!/b^{2k} \geq (k/2)^k/(\sqrt{k}/2)^{2k} \geq 2^k ,$$

thus the condition holds. Therefore, $\tilde{p}'(z) > 0$ for $z < -b$ and $p(z)$ must be monotone increasing for $z < -b$. Then combined with the condition $p(-b) < 0$, which is implied from the previous case ($z \in [-b, b]$), we have $p(z) < 0$ for $z < -b$.

For the Property 2, we show that $p(z) = 2^{-\Theta(k)}$ for $z \in [0, b/2]$, $z \in [-b/2, 0]$, $z \in [-b, -b/2]$ and $z \in [-\infty, -b]$ separately. For $z \in [0, b/2]$ notice that $p(z)$ is convex for $z \in [0, b]$; therefore, it suffices to show $p(0) = 2^{-\Theta(k)}$. Note that $p(0) = \frac{q(b)}{r(b)}(2k-1)!!/\sqrt{\mathbf{E}_{u \sim \mathcal{N}_1} \tilde{p}^2(u)}$, and we have $\left|\frac{q(b)}{r(b)}\right| \geq b^{3k}/\max(b^{2k}, (2k-1)!!) \geq \Omega(k)^{k/2}$. Therefore, since $\mathbf{E}_{u \sim \mathcal{N}_1}[\tilde{p}^2(u)] \leq (O(k))^{3k}$, we have $p(0) \geq 2^{-O(k)}$. This completes the proof of the case $z \in [0, b/2]$. For the case $z \in [-b/2, 0]$, it immediately follows from the fact that $|p(-z)| \geq |p(z)|$ by the definition. For the case $z \in [-b, -b/2]$, we have $p(z) \leq (p(b) - (b/2)^{3k})/\sqrt{\mathbf{E}_{u \sim \mathcal{N}_1} \tilde{p}^2(u)} \leq -2^{-O(k)}$. Then, the case $z \in [-\infty, -b]$ immediately follows from the fact that $p(z)$ is monotone increasing in this interval. This completes the proof of Property 2. $\square$

Given Claim B.5, we let $p$ be the polynomial in Claim B.5 with $b = -\max(2|t^*|, 4)$. Then, given that $D$ satisfies the reliability condition with respect to $f$, it is easy to see the polynomial $p(\langle \mathbf{w}^*, \mathbf{x} \rangle)$ satisfies the requirements. This completes the proof of Lemma 2.2.

## B.5   Proof of Lemma 2.4

We start by bounding the Frobenius norm of $\mathbf{T}^m$, i.e., $\|\mathbf{T}^m\|_F$. Notice that by taking $p : \mathbb{R} \to \mathbb{R}$ to be any degree-$m$ polynomial such that $\mathbf{E}_{\mathbf{x} \sim \mathcal{N}_d}[p(\mathbf{x})] = 0$ and $\mathbf{E}_{\mathbf{x} \sim \mathcal{N}_d}[p(\mathbf{x})^2] = 1$, then $\|\mathbf{T}^m\|_F = \max_p \mathbf{E}_{(\mathbf{x}, y) \sim D}[\mathbf{1}(y = -1)p(\mathbf{x})]$. We leverage the following standard fact about the concentration of polynomials lemma known as Bonami-Beckner inequality or simply Gaussian hypercontractivity.

**Fact B.6** (Hypercontractivity Concentration Inequality [Bog98, Nel73])**.** *Consider any $m$-degree, unit variance polynomial $p : \mathbb{R}^d \to \mathbb{R}$ with respect to $\mathcal{N}_d$. Then, for any $\lambda \geq 0$*

$$\Pr_{\mathbf{x} \sim \mathcal{N}_d} \left[ \left| p(\mathbf{x}) - \mathbf{E}_{\mathbf{u}}[p(\mathbf{u})] \right| \geq \lambda \right] \leq e^2 e^{-\left(c\lambda^2\right)^{1/m}} ,$$

*where $c$ is an absolute constant.*

Using Fact B.6, we have that for any zero mean and unit variance polynomial $p$,

$$\mathbf{E}_{(\mathbf{x},y)\sim D}[\mathbf{1}(y=-1)p(\mathbf{x})] \leq \mathbf{E}_{(\mathbf{x},y)\sim D}[\mathbf{1}(y=-1)|p(\mathbf{x})|] \leq \int_0^\infty \min\left(\gamma, e^2 e^{-\left(c\lambda^2\right)^{1/m}}\right) d\lambda$$

$$= O\left(\gamma \log(1/\gamma)^{m/2}\right) .$$

Therefore, $\|\mathbf{T}^m\|_F = O\left(\gamma \log(1/\gamma)^{m/2}\right)$.

To prove the Property 1, since $\|\mathbf{T}'^m\|_F \leq \|\mathbf{T}^m\|_F + \tau/(4\sqrt{k}) = O\left(\gamma \log(1/\gamma)^{m/2}\right) + \tau/(4\sqrt{k})$, there can be at most $\|\mathbf{T}^m\|_F^2 / \left(\tau/\sqrt{k}\right)^2 = O(\gamma^2 \log(1/\gamma)^m k/\tau^2 + 1)$ many singular vectors with singular values greater than $\tau/(4\sqrt{k})$. Therefore, $\dim(V)$ is at most $O(\gamma^2 \log(1/\gamma)^m k/\tau^2 + 1)$.

For the Property 2, suppose $\|\mathrm{proj}_V(\mathbf{v}^*)\|_2 = o\left(\tau/\left(\sqrt{k}\gamma \log(1/\gamma)^{k/2}\right)\right)$, then we have for any $m$,

$$\|\mathbf{v}^{*\top}\mathbf{T}'^m\|_2 \leq \|\mathbf{v}^{*\top}\mathbf{T}^m\|_2 + \tau/(4\sqrt{k})$$

$$\leq \|\mathbf{T}^m\|_F \|\mathrm{proj}_V(\mathbf{v}^*)\|_2 + \tau/(4\sqrt{k})\|\mathrm{proj}_{\perp V}(\mathbf{v}^*)\|_2 + \tau/(4\sqrt{k}) \leq (5/8)\tau/\sqrt{k} .$$

However, since $\mathbf{E}_{(\mathbf{x},y)\sim D}[\mathbf{1}(y=-1)p(\langle\mathbf{v}^*,\mathbf{x}\rangle)] = \sum_{m=1}^k a_i \mathbf{v}^{*\top}\mathbf{T}^m\mathbf{v}^{*\otimes m-1} \geq \tau$ for some $a_1^2 + \cdots + a_k^2 = 1$, we know there must be an $m$ such that

$$\left|\mathbf{v}^{*\top}\mathbf{T}^m\mathbf{v}^{*\otimes m-1}\right| \geq \tau/\sqrt{k} .$$

Therefore, we can write

$$\|\mathbf{v}^{*\top}\mathbf{T}'^m\|_2 \geq \|\mathbf{v}^{*\top}\mathbf{T}^m\|_2 - \tau/(4\sqrt{k}) \geq \tau/\sqrt{k} - \tau/(4\sqrt{k}) = (3/4)\tau/\sqrt{k} ,$$

which contradicts $\|\mathbf{v}^{*\top}\mathbf{T}'^m\|_2 \leq (5/8)\tau/\sqrt{k}$. This completes the proof.

## B.6    Proof of Proposition 2.1

The pseudocode for the algorithm establishing Proposition 2.1 is given in Algorithm 2.

We can now prove Proposition 2.1.

*Proof of Proposition 2.1.* We first introduce the following fact about learning Chow tensors.

**Fact B.7.** *Fix $m \in \mathbb{Z}_+$, and $\epsilon, \delta \in (0,1)$. Let $D$ be a distribution on $\mathbb{R}^d \times [\pm 1]$ with standard normal marginals. There is an algorithm that with $N = d^{O(m)} \log(1/\delta)/\epsilon^2$ samples and $\mathrm{poly}(d, N)$ runtime, outputs an approximation $\mathbf{T}'^m$ of the order-$m$ Chow-parameter tensor $\mathbf{T}^m = \mathbf{E}_{(\mathbf{x},y)\sim D}[y\mathbf{H}^m(\mathbf{x})]$ such that with probability $1-\delta$, it holds $\|\mathbf{T}'^m - \mathbf{T}^m\|_F \leq \epsilon$.*

*Proof.* Let $\mathbf{T}'^m$ be the empirical estimation of $\mathbf{T}^m$ using $N = d^{cm} \log(1/\delta)/\epsilon^2$ for sufficiently large universal constant $c$. We start by showing $\mathbf{E}_{(\mathbf{x},y)\sim D}\left[\|y\mathbf{H}^m(\mathbf{x})\|_F^2\right] \leq d^m$. Notice that,

$$\mathbf{E}_{(\mathbf{x},y)\sim D}\left[\|y\mathbf{H}^m(\mathbf{x})\|_F^2\right] = \mathbf{E}_{(\mathbf{x},y)\sim D}\left[y^2\|\mathbf{H}^m(\mathbf{x})\|_F^2\right] \leq \mathbf{E}_{(\mathbf{x},y)\sim D}\left[\|\mathbf{H}^m(\mathbf{x})\|_F^2\right] = \mathbf{E}_{\mathbf{x}\sim\mathbb{R}^d}\left[\|\mathbf{H}^m(\mathbf{x})\|_F^2\right] .$$

Notice that each entry of $\mathbf{H}^m(\mathbf{x})$ must be some $\alpha h(\mathbf{x})$ where $\alpha \in [0,1]$ and $h(\mathbf{x})$ is a unit variance Hermite polynomial. Therefore, $\mathbf{E}_{(\mathbf{x},y)\sim D}\left[\|\mathbf{H}^m(\mathbf{x})\|_F^2\right] \leq d^m$ and thus $\mathbf{E}_{(\mathbf{x},y)\sim D}\left[\|y\mathbf{H}^m(\mathbf{x})\|_F^2\right] \leq d^m$.

Using the above, we have

$$\mathbf{E}_{(\mathbf{x},y)\sim D}\left[\|y\mathbf{H}^m(\mathbf{x}) - \mathbf{T}^m\|_F^2\right] = \mathbf{E}_{(\mathbf{x},y)\sim D}\left[\|y\mathbf{H}^m(\mathbf{x})\|_F^2\right] - \|\mathbf{T}^m\|_F^2 \leq d^m .$$

Then, applying Chebyshev's inequality gives $\|\mathbf{T}'^m - \mathbf{T}^m\|_F \leq \epsilon$. $\qquad\square$

**Input:** Sample access to a joint distribution $D$ of $(\mathbf{x}, y)$ supported on $\mathbb{R}^d \times \{\pm 1\}$ with the marginal $D_{\mathbf{x}} = \mathcal{N}_d$. Let $\epsilon, \delta \in (0, 1)$ and suppose $D$ satisfies the reliability condition with respect to an LTF $f(\mathbf{x}) = \text{sign}(\langle \mathbf{w}^*, \mathbf{x} \rangle - t^*)$ with $t^* = O\left(\sqrt{\log(1/\epsilon)}\right)$ and $\mathbf{Pr}_{(\mathbf{x}, y) \sim D}[y = -1] \geq \epsilon$.

**Output:** With probability at least $\Omega(1)$, the algorithm outputs a unit vector $\mathbf{v}$ such that $\langle \mathbf{v}, \mathbf{w}^* \rangle = \max\left(\log(1/\epsilon)^{-O(t^{*2})}, \epsilon^{O(1)}\right)$.

1. Let $c_1$ be a sufficiently large universal constant and $c_2$ be a sufficiently large universal constant depending on $c_1$. Let $S$ be a set of $N = d^{c_2 \max(t^{*2}, 1)} \log(1/\delta)/\epsilon^2$ many samples from $D$. For $m = 1, \cdots, c_1 t^{*2}$, with 1/2 probability, take

$$\mathbf{T}'^m = \underset{(\mathbf{x}, y) \sim_u S}{\mathbf{E}}[\mathbf{1}(y = -1)\mathbf{H}^m(\mathbf{x})],$$

   to be the empirical Chow-tensor on negative samples. Otherwise, take

$$\mathbf{T}'^m = \underset{(\mathbf{x}, y) \sim_u S}{\mathbf{E}}[y\mathbf{H}^m(\mathbf{x})],$$

   to be the empirical Chow-tensor on all samples. Let $\gamma$ be the empirical estimation of $\mathbf{Pr}_{(\mathbf{x}, y) \sim D}[y = -1]$ with error at most $\epsilon/100$ with a sufficiently small constant failure probability.

2. Take $\tilde{\mathbf{T}}'^m \in \mathbb{R}^{d \times d^{m-1}}$ to be the flattened version of $\mathbf{T}'^m$, and let $V_m$ be the subspace spanned by the left singular vectors of $\tilde{\mathbf{T}}'^m$ whose singular values are greater than $2^{-ct^{*2}} \gamma$ where $c$ is a sufficiently large constant. Let $V$ be the union of all $V_m$ and output $\mathbf{v}$ to be a random unit vector chosen from $V$.

**Algorithm 2:** Finding a Direction with High Correlation.

We prove Proposition 2.1 for two cases: (a) $\max(\log(1/\epsilon)^{-O(t^{*2})}, \epsilon^{O(1)}) = \log(1/\epsilon)^{-O(t^{*2})}$; and (b) $\max(\log(1/\epsilon)^{-O(t^{*2})}, \epsilon^{O(1)}) = \epsilon^{O(1)}$. We first prove Case (a) as the other case is similar. For Case (a), it suffices for us to prove that $\langle \mathbf{v}, \mathbf{w}^* \rangle = \log(1/\epsilon)^{-O(t^{*2})}$ happens with $\Omega(1)$ probability given the algorithm chooses $\mathbf{T}'^m = \mathbf{E}_{(\mathbf{x}, y) \sim_u S}[\mathbf{1}(y = -1)\mathbf{H}^m(\mathbf{x})]$ (which happens with 1/2 probability). Let $k = c_1 t^{*2}$ and $\tau/(4\sqrt{k}) = 2^{-ct^{*2}} \gamma$. By Fact B.7, we have $\|\mathbf{T}'^m - \mathbf{T}^m\|_F \leq \tau/(4\sqrt{k})$. Then from Lemma 2.2 and Lemma 2.4, if we take $\mathbf{v}$ to be a random vector in $V$, we will with constant probability have,

$$\langle \mathbf{v}, \mathbf{w}^* \rangle = \frac{\Omega\left(\tau/\left(\sqrt{k}\gamma \log(1/\gamma)^{k/2}\right)\right)}{\dim(V)^{1/2}} = \Omega\left(\tau^2/\left(\log(1/\gamma)^k k^{3/2} \gamma^2\right)\right)$$
$$= \Omega\left(\log(1/\gamma)^{-O(t^{*2})} k^{-3/2}\right).$$

Since $\mathbf{Pr}_{(\mathbf{x}, y) \sim D}[y = -1] \geq \epsilon$, we have $\gamma = \Omega(\epsilon)$. Also, considering $k = O(\max\{t^{*2}, 1\}) = O(\log(1/\epsilon))$, we get $\langle \mathbf{v}, \mathbf{w}^* \rangle = \log(1/\epsilon)^{-O(t^{*2})}$.

For Case (b), the proof remains the same except we take $\mathbf{T}'^m = \mathbf{E}_{(\mathbf{x}, y) \sim_u S}[y\mathbf{H}^m(\mathbf{x})]$ and use fact below instead of Lemma 2.4.

**Fact B.8** (Lemma 5.10 in [DKK$^+$22])**.** *Let $D$ be the joint distribution of $(\mathbf{x}, y)$ supported on $\mathbb{R}^d \times \{\pm 1\}$ with the marginal $D_{\mathbf{x}} = \mathcal{N}_d$. Let $p : \mathbb{R} \mapsto \mathbb{R}$ be a univariate, mean zero, unit variance polynomial of degree $k$ such that for some unit vector $\mathbf{v}^* \in \mathbb{R}^d$ it holds $\mathbf{E}_{(\mathbf{x}, y) \sim D}[yp(\langle \mathbf{v}^*, \mathbf{x} \rangle)] \geq \tau$ for some $\tau \in (0, 1]$. Let $\mathbf{T}'^m$ be an approximation of the order-$m$ Chow-parameter tensor $\mathbf{T}^m = \mathbf{E}_{(\mathbf{x}, y) \sim D}[y\mathbf{H}_m(\mathbf{x})]$ such that $\|\mathbf{T}'^m - \mathbf{T}^m\|_F \leq \tau/(4\sqrt{k})$. Denote by $V_m$ the subspace spanned by the left singular vectors of flattened $\mathbf{T}'^m$ whose singular values are greater than $\tau/(4\sqrt{k})$. Moreover, denote by $V$ the union of $V_1, \cdots, V_k$. Then we have that*

*1. $\dim(V) \leq 4k/\tau^2$, and*

2. $\|\text{proj}_V(\mathbf{v}^*)\|_2 \geq \tau / \left(4\sqrt{k}\right)$.

Then the same argument will give $\langle \mathbf{v}, \mathbf{w}^* \rangle = \epsilon^{O(1)}$. The above described algorithm uses at most $N = d^{O(t^{*2})}/\epsilon^2$ samples and $\text{poly}(N, 1/\epsilon)$ runtime. This completes the proof of Proposition 2.1. $\qquad\square$

## B.7 Proof of Lemma 2.5

Item 1 follows from the definition of $D'$ and the fact that $D$ has marginal distribution $D_{\mathbf{x}} = \mathcal{N}_d$.

For proving Item 2, we consider two cases: The first case is when $t^* > 0$ and the second one when $t^* \leq 0$. For the case $t^* \leq 0$, we prove that the distribution $D'$ satisfies the reliability condition with respect to $h'(\mathbf{x}) = \text{sign}(\langle \mathbf{w}', \mathbf{x} \rangle)$ where $\mathbf{w}' = \mathbf{w}^{*\perp\mathbf{w}}/\|\mathbf{w}^{*\perp\mathbf{w}}\|_2$. Notice that for any $(\mathbf{x}, y)$ such that $\mathbf{x} \in B$ and $h'(\mathbf{x}^{\perp\mathbf{w}}) = 1$, we have

$$
\begin{aligned}
f(\mathbf{x}) &= \text{sign}(\langle \mathbf{w}^*, \mathbf{x} \rangle - t^*) \\
&= \text{sign}(\langle \text{proj}_{\perp\mathbf{w}}(\mathbf{w}^*), \mathbf{x} \rangle + \langle \text{proj}_{\mathbf{w}}(\mathbf{w}^*), \mathbf{x} \rangle - t^*) \\
&= \text{sign}(\langle \mathbf{w}^{*\perp\mathbf{w}}, \mathbf{x}^{\perp\mathbf{w}} \rangle + \langle \text{proj}_{\mathbf{w}}(\mathbf{w}^*), \mathbf{x} \rangle - t^*) ,
\end{aligned}
$$

where $\langle \mathbf{w}^{*\perp\mathbf{w}}, \mathbf{x}^{\perp\mathbf{w}} \rangle \geq 0$ since $h'(\mathbf{x}) = \text{sign}(\langle \mathbf{w}^{*\perp\mathbf{w}}, \mathbf{x}^{\perp\mathbf{w}} \rangle/\|\mathbf{w}^{*\perp\mathbf{w}}\|_2) \geq 0$. Then we have

$$
\begin{aligned}
f(\mathbf{x}) &\geq \text{sign}(\langle \text{proj}_{\mathbf{w}}(\mathbf{w}^*), \mathbf{x} \rangle - t^*) \\
&= \text{sign}(\langle \text{proj}_{\mathbf{w}}(\mathbf{w}^*)/\|\text{proj}_{\mathbf{w}}(\mathbf{w}^*)\|_2, \mathbf{x} \rangle - t^*/\|\text{proj}_{\mathbf{w}}(\mathbf{w}^*)\|_2 \\
&\geq \text{sign}(\langle \mathbf{w}, \mathbf{x} \rangle - t^*) \\
&\geq \text{sign}(\langle \mathbf{w}, \mathbf{x} \rangle - t) \\
&= h(\mathbf{x}) = +1 ,
\end{aligned}
\tag{2}
$$

where Inequality (2) follows from the fact that $\langle \mathbf{w}, \mathbf{w}^* \rangle > 0$, $\|\text{proj}_{\mathbf{w}}(\mathbf{w}^*)\|_2 \in (0, 1]$ and $t^* \leq 0$. Since $D$ satisfies the reliability condition with respect to $f$, we have that it must be the case $y = +1$. Therefore, $D'$ satisfies the reliability condition with respect to $h'$.

For the case $t^* > 0$, we prove that the distribution $D'$ satisfies the reliability condition with respect to $h'(\mathbf{x}) = \text{sign}(\langle \mathbf{w}', \mathbf{x} \rangle - t')$ where $\mathbf{w}' = \mathbf{w}^{*\perp\mathbf{w}}/\|\mathbf{w}^{*\perp\mathbf{w}}\|_2$ and $t' = t^*$. Notice that for any $(\mathbf{x}, y)$ such that $\mathbf{x} \in B$ and $h'(\mathbf{x}^{\perp\mathbf{w}}) = 1$, we have

$$
\begin{aligned}
f(\mathbf{x}) &= \text{sign}(\langle \mathbf{w}^*, \mathbf{x} \rangle - t^*) \\
&= \text{sign}(\langle \text{proj}_{\perp\mathbf{w}}\mathbf{w}^*, \mathbf{x} \rangle + \langle \text{proj}_{\mathbf{w}}\mathbf{w}^*, \mathbf{x} \rangle - t^*) \\
&= \text{sign}\left(\|\mathbf{w}^{*\perp\mathbf{w}}\|_2 \langle \mathbf{w}', \mathbf{x} \rangle - \|\mathbf{w}^{*\perp\mathbf{w}}\|_2 t^* + \langle \text{proj}_{\mathbf{w}}\mathbf{w}^*, \mathbf{x} \rangle - (1 - \|\mathbf{w}^{*\perp\mathbf{w}}\|_2)t^*\right) ,
\end{aligned}
$$

where $\|\mathbf{w}^{*\perp\mathbf{w}}\|_2 \langle \mathbf{w}', \mathbf{x} \rangle - \|\mathbf{w}^{*\perp\mathbf{w}}\|_2 t^* \geq 0$ since $h'(\mathbf{x}) = +1$. Then we have

$$
\begin{aligned}
f(\mathbf{x}) &\geq \text{sign}(\langle \text{proj}_{\mathbf{w}}\mathbf{w}^*, \mathbf{x} \rangle - (1 - \|\mathbf{w}^{*\perp\mathbf{w}}\|_2)t^*) \\
&= \text{sign}\left(\sqrt{1 - \|\mathbf{w}^{*\perp\mathbf{w}}\|_2^2}\, \langle \mathbf{w}, \mathbf{x} \rangle - (1 - \|\mathbf{w}^{*\perp\mathbf{w}}\|_2)t^*\right) \\
&\geq \text{sign}\left(\sqrt{1 - \|\mathbf{w}^{*\perp\mathbf{w}}\|_2^2}\, (\langle \mathbf{w}, \mathbf{x} \rangle - t^*)\right) = h(\mathbf{x}) = 1 ,
\end{aligned}
$$

where the second equation follows from $\langle \mathbf{w}, \mathbf{w}^* \rangle > 0$ and the third one follows from $t^* > 0$. Since $D$ satisfies the reliability condition with respect to $f$, we have that it must be the case $y = +1$. Therefore, $D'$ satisfies the reliability condition with respect to $h'$.

For proving Item 3, notice that $R_+(h; D) \geq \epsilon/2$ implies

$$
\Pr_{(\mathbf{x}', y) \sim D'}[y = -1] = R_+(h; D)/\Pr_{(\mathbf{x}, y) \sim D}(\mathbf{x} \in B) \geq \epsilon/2 .
$$

This completes the proof.

## B.8 Proof of Theorem B.4

We first give the algorithm in Theorem B.4, which is a more detailed version of Algorithm 1.

---

**Input:** $\epsilon \in (0,1)$, $\alpha \in (0,1/2)$ and sample access to a joint distribution $D$ of $(\mathbf{x}, y)$ supported on $\mathbb{R}^d \times \{\pm 1\}$ with $\mathbf{x}$-marginal $D_{\mathbf{x}} = \mathcal{N}_d$.
**Output:** $h(\mathbf{x}) = \text{sign}(\langle \mathbf{w}, \mathbf{x} \rangle - t)$ that is $\epsilon$-reliable with respect to the class $\mathcal{H}_d^\alpha$.

1. Check if $\mathbf{Pr}_{(\mathbf{x},y)\sim D}[y = -1] \leq \epsilon/2$ (with sufficiently small constant failure probability). If so, return the $+1$ constant hypothesis. If the parameters satisfy

$$\min\left(2^{\log(1/\epsilon)^{O(\log(1/\alpha))}}, 2^{\text{poly}(1/\epsilon)}\right) = 2^{\log(1/\epsilon)^{O(\log(1/\alpha))}} \; ,$$

   then set the correlation parameter $\zeta = \log(1/\epsilon)^{-ct^{*2}}$, where $c$ is a sufficiently large universal constant. Otherwise, set $\zeta = \epsilon^c$ for a sufficiently large universal constant $c$. For $t = -\Phi^{-1}(\alpha)$ and $t = \Phi^{-1}(\alpha)$, do Steps (2) and (3).

2. Initialize $\mathbf{w}$ to be a random unit vector in $\mathbb{R}^d$. Let the update step size $\lambda = \zeta$ and repeat the following process until $\lambda \leq \epsilon/100$.

   (a) Use samples from $D$ to check if the hypothesis $h(\mathbf{x}) = \text{sign}(\langle \mathbf{w}, \mathbf{x} \rangle - t)$ satisfies $R_+(h; D) \leq \epsilon/2$. If so, go to Step (3).

   (b) With $1/2$ probability, let $\mathbf{w} = -\mathbf{w}$. Let $B = \{\mathbf{x} \in \mathbb{R}^d : \langle \mathbf{w}, \mathbf{x} \rangle - t \geq 0\}$, and let $D'$ be the distribution of $(\mathbf{x}^{\perp \mathbf{w}}, y)$ for $(\mathbf{x}, y) \sim D$ given $\mathbf{x} \in B$. Use the algorithm of Proposition 2.1 on $D'$ to find a unit vector $\mathbf{v}$ such that $\langle \mathbf{v}, \mathbf{w} \rangle = 0$ and $\left\langle \mathbf{v}, \frac{\text{proj}_{\perp \mathbf{w}}(\mathbf{w}^*)}{\|\text{proj}_{\perp \mathbf{w}}(\mathbf{w}^*)\|_2} \right\rangle \geq \zeta$.
   Then, update $\mathbf{w}$ as follows: $\mathbf{w}_{\text{update}} = \frac{\mathbf{w} + \lambda \mathbf{v}}{\|\mathbf{w} + \lambda \mathbf{v}\|_2}$ .

   (c) Repeat Steps (2a) and (2b) $c/\zeta^2$ times, where $c$ is a sufficiently large universal constant, with the same step size $\lambda$. After that, update the new step size as $\lambda_{\text{update}} = \lambda/2$.

3. Check if $h(\mathbf{x}) = \text{sign}(\langle \mathbf{w}, \mathbf{x} \rangle - t)$ satisfies $R_+(h; D) \leq \epsilon/2$. If so, add it to the set $S$. For each choice of value $t$ in Step (1), repeat Step (2) $2^{1/\zeta^c}$ many times where $c$ is a sufficiently large constant.

4. Let $S$ be the set in Step (3). Return the hypothesis $h(\mathbf{x}) = \text{sign}(\langle \mathbf{w}, \mathbf{x} \rangle - t)$, where $(\mathbf{w}, t) = \text{argmin}_{(\mathbf{w},t) \in S} t$. Return the $-1$ constant hypothesis if $S$ is empty.

---

**Algorithm 3:** Reliably Learning General Halfspaces with Gaussian Marginals (detailed version of Algorithm 1).

*Proof of Theorem B.4.* Without loss of generality, we assume $\epsilon > \alpha$ for the following reason. Suppose $\alpha \leq \epsilon$, then we can simply return one of the constant hypotheses that is closest to $f$. Let $f(\mathbf{x}) = \text{sign}(\langle \mathbf{w}^*, \mathbf{x} \rangle - t^*)$ be the optimal LTF with $\alpha$ bias, namely,

$$f = \underset{f \in \mathcal{H}_d^\alpha \wedge R_+(h;D)=0}{\text{argmin}} R_-(f; D) \; .$$

We prove that with high probability, Algorithm 3 returns a hypothesis $h(\mathbf{x}) = \text{sign}(\langle \mathbf{w}, \mathbf{x} \rangle - t)$ such that $R_+(h; D) \leq \epsilon$ and $R_-(h; D) \leq R_-(f; D) + \epsilon$. Notice that at some point, we will run Step (2) with $t = t^*$. We show that given $t = t^*$, Step (2) will with probability $2^{-\text{poly}(1/\zeta)}$ returns a $h$ with $R_+(h; D) \leq \epsilon/2$. For convenience, we assume $h$ never satisfies $R_+(h; D) \leq \epsilon/2$ in Step (2a) (otherwise, we are done). Furthermore, we assume the subroutine algorithm in Proposition 2.1 used in Step (2b) always succeed, and we always have $\langle \mathbf{w}, \mathbf{w}^* \rangle \geq 0$, since both happen with constant probability and we are running the update at most $O(\log(1/\epsilon)/\zeta^2) = \text{poly}(1/\zeta)$ many times.

Let $\eta = \|\text{proj}_{\perp \mathbf{w}} \mathbf{w}^*\|_2$. Now, we will prove by induction that each time after $c/\zeta^2$ many updates in step (2b), we always have $\|\text{proj}_{\perp \mathbf{w}} \mathbf{w}^*\|_2 \leq 3\lambda$ (without loss of generality, we assume $\lambda$ is always at most a sufficiently small constant). Notice that by Lemma 2.5 and Proposition 2.1, we have that at each update, given the subroutine algorithm in Proposition 2.1 succeeds, the update direction $\mathbf{v}$ always satisfies $\left\langle \mathbf{v}, \frac{\text{proj}_{\perp \mathbf{w}}(\mathbf{w}^*)}{\eta} \right\rangle \geq \zeta$, which implies $\langle \mathbf{v}, \text{proj}_{\perp \mathbf{w}}(\mathbf{w}^*) \rangle \geq \eta \zeta$. Then we have $\langle \mathbf{w}, \mathbf{w}^* \rangle = \sqrt{1 - \eta^2} \geq 1 - \eta^2$. When we update $\mathbf{w}$, there are two possibilities. Either $\lambda > \eta \zeta/2$ or

$\lambda \leq \eta\zeta/2$. If $\lambda \leq \eta\zeta/2$, using Fact 2.6, we will have $\langle \mathbf{w}_{\text{update}}, \mathbf{w}^* \rangle \geq \langle \mathbf{w}, \mathbf{w}^* \rangle + \lambda^2/2$. If $\lambda > \eta\zeta/2$, we have

$$\left\| \text{proj}_{\perp \mathbf{w}_{\text{update}}}(\mathbf{w}^*) \right\|_2 = \left\| \text{proj}_{\perp \mathbf{w}^*}(\mathbf{w}_{\text{update}}) \right\|_2 \leq \left\| \text{proj}_{\perp \mathbf{w}^*}(\mathbf{w} + \lambda \mathbf{v}) \right\|_2$$
$$\leq \left\| \text{proj}_{\perp \mathbf{w}^*}(\mathbf{w}) \right\|_2 + \lambda \leq 3\lambda \ .$$

For the base case, when we do the first group of $c/\zeta^2$ many updates, combining the two cases above and the fact that $\lambda = \zeta \geq \eta\zeta$ gives that after $c/\zeta^2$ many updates, we must have $\|\text{proj}_{\perp \mathbf{w}}\mathbf{w}^*\|_2 \leq 3\lambda$. For the induction case, given $\eta \leq 6\lambda$ (which follows from the previous group of $c/\zeta^2$ many updates), combining the two cases above gives that after $c/\zeta^2$ many updates, we must have $\|\text{proj}_{\perp \mathbf{w}}\mathbf{w}^*\|_2 \leq 3\lambda$. This proves that after each group of $c/\zeta^2$ many updates, we have $\|\text{proj}_{\perp \mathbf{w}}\mathbf{w}^*\|_2 \leq 3\lambda$.

Therefore, when we have $\lambda \leq \epsilon/100$, we get $\|\text{proj}_{\perp \mathbf{w}}\mathbf{w}^*\|_2 \leq 3\epsilon/100$, which implies $R_+(h; D) \leq \epsilon/2$ since $D$ satisfies the reliability condition with respect to $f$. Given $R_+(h; D) \leq \epsilon/2$, using the fact that $t - t^* \leq 0$ (since we always output the hypothesis with the smallest $t$) implies $\mathbf{Pr}_{(\mathbf{x},y)\sim D}[h(\mathbf{x}) = 1] \geq \mathbf{Pr}_{(\mathbf{x},y)\sim D}[f(\mathbf{x}) = 1]$, we have that

$$R_-(h; D) = \Pr_{(\mathbf{x},y)\sim D}[y = 1] - \Pr_{(\mathbf{x},y)\sim D}[y = 1 \wedge h(\mathbf{x}) = 1]$$
$$= \Pr_{(\mathbf{x},y)\sim D}[y = 1] - \Pr_{(\mathbf{x},y)\sim D}[h(\mathbf{x}) = 1] + R_+(h; D)$$
$$\leq \Pr_{(\mathbf{x},y)\sim D}[y = 1] - \Pr_{(\mathbf{x},y)\sim D}[f(\mathbf{x}) = 1] + \epsilon = R_-(f; D) + \epsilon \ .$$

This completes the proof. $\qquad\qquad\qquad\qquad\qquad\qquad\qquad\qquad\qquad\qquad\qquad\qquad$ $\square$

## C  Omitted Proofs from Section 3

We first restate our main SQ hardness result.

**Theorem C.1** (SQ Lower Bound for Reliable Learning). *For any $\alpha > 0$ that is at most a sufficiently small absolute constant and $\epsilon \in (0, \alpha/3)$, any SQ algorithm that reliably learns $\alpha$-biased LTFs (for positive labels) on $\mathbb{R}^d$ under Gaussian marginals to additive error $\epsilon$ either (i) requires at least one query of tolerance at most $d^{-\Omega\left(\log\left(\frac{1}{\alpha}\right)\right)}$, or (ii) requires at least $2^{d^{\Omega(1)}}$ queries.*

### C.1  Proof of Lemma 3.3

We let $P_n$ denote the set of all polynomials $p : \mathbb{R} \to \mathbb{R}$ of degree at most $n$ and let $L^1_+(\mathbb{R})$ denote the set of all nonnegative functions in $L^1(\mathbb{R})$. First, we write the conditions as the primal linear program.

$$\begin{aligned} \text{find} \qquad & g \in L^1(\mathbb{R}) \\ \text{such that} \quad & \mathbf{E}_{z\sim\mathcal{N}_1}[p(z)g(z)] = 0 \ , \quad \forall p \in P_n \\ & g(z) \geq 1 \ , \quad \forall z \geq c \\ & \|g\|_\infty \leq 1 \end{aligned}$$

By introducing variables $H \in L^1(R)$ and $h \in L^1_+(R)$, we rewrite the primal LP as

$$\begin{aligned} \text{find} \qquad\qquad & g \in L^1(\mathbb{R}) \\ \text{such that} \qquad\quad & \mathbf{E}_{z\sim\mathcal{N}_1}[p(z)g(z)] = 0 \ , & \forall p \in P_n \\ \mathbf{E}_{z\sim\mathcal{N}_1}[g(z)h(z)\mathbb{1}\{z \geq c\}] \geq & \|h(z)\mathbb{1}\{z \geq c\}\|_1 \ , & \forall h \in L^1_+(\mathbb{R}) \\ \mathbf{E}_{z\sim\mathcal{N}_1}[g(z)H(z)] \leq & \|H\|_1 \ , & \forall H \in L^1(\mathbb{R}) \end{aligned}$$

The following fact is an application of (an infinite generalization of) LP duality:

**Claim C.2.** *The LP above is feasible if there exists no polynomial $p$ of degree $n$ such that*

$$\mathop{\mathbf{E}}_{z \sim \mathcal{N}_1}[|p(z)|\mathbb{1}(z \leq c)] < \mathop{\mathbf{E}}_{z \sim \mathcal{N}_1}[p(z)\mathbb{1}(z \geq c)] \ .$$

*Proof of Claim C.2.* First, we introduce some notation. We use $(\tilde{h}, c)$ for the inequality $\mathbf{E}_{z \sim \mathcal{N}}[\beta(z)\tilde{h}(z)] + c \leq 0$, where $\tilde{h} \in L^1(\mathbb{R})$ and $c \in \mathbb{R}$. Moreover, let $\mathcal{S}$ be the set that contains all such tuples that describe the target system. For the set $\mathcal{S}$, the closed convex cone over $L^1(\mathbb{R}) \times \mathbb{R}$ is the smallest closed set $\mathcal{S}_+$ satisfying the following: (a) if $A \in \mathcal{S}_+$ and $B \in \mathcal{S}_+$ then $A + B \in \mathcal{S}_+$; and (b) if $A \in \mathcal{S}_+$ then $\lambda A \in \mathcal{S}_+$ for all $\lambda \geq 0$. Note that the $\mathcal{S}_+$ contains the same feasible solutions as $\mathcal{S}$. In order to prove the statement, we need the following LP duality from [Fan68].

**Fact C.3** (Theorem 1 of [Fan68])**.** *If $\mathcal{X}$ is a locally convex, real separated vector space. Then, a linear system described by $\mathcal{S}$ is feasible (i.e., there exists a $g \in \mathcal{X}^*$) if and only if $(0, 1) \notin \mathcal{S}_+$.*

Our dual LP is defined by the following inequalities:

(a) $(p, 0)$ for $p \in \mathcal{P}_n$;

(b) $(-\tilde{h}, -\|\tilde{h}\|_1)$ for all $\tilde{h}(z) = h(z)\mathbb{1}\{z \geq c\}$, where $h \in L^1_+(\mathbb{R})$; and

(c) and $(H, -\|H\|_1)$ for all $H \in L^1(\mathbb{R})$.

Furthermore, the LP is also equivalent to breaking the last inequality into two, i.e., $(\tilde{H}_1, -\|\tilde{H}_1\|_1)$ for all $\tilde{H}_1(z) = H_1(z)\mathbb{1}\{z < c\}$, where $H_1 \in L^1(\mathbb{R})$, and $(\tilde{H}_2, -\|\tilde{H}_2\|_1)$ for all $\tilde{H}_2(z) = H_2(z)\mathbb{1}\{z \geq c\}$ where $H_2 \in L^1_+(\mathbb{R})$. By taking the convex cone defined from the above inequalities, we get the following set

$$\mathcal{S}_+ = \Big\{ \Big( p - \tilde{h} + \tilde{H}_1 + \tilde{H}_2, \|\tilde{h}\|_1 - \|\tilde{H}_1 + \tilde{H}_2\|_1 \Big) \ \Big| \ p \in \mathcal{P}_n,$$
$$\tilde{h}(z) = h(z)\mathbb{1}\{z \geq c\} \text{ for } h \in L^1_+(\mathbb{R}),$$
$$\tilde{H}_1(z) = H_1(z)\mathbb{1}\{z < c\} \text{ for } H_1 \in L^1(\mathbb{R}),$$
$$\tilde{H}_2(z) = H_2(z)\mathbb{1}\{z \geq c\} \text{ for } H_2 \in L^1_+(\mathbb{R}) \Big\} \ .$$

We first show that $\mathcal{S}_+$ is closed. To do so, we prove that $\mathcal{S}_+$ is closed under limits. First, we assume, in order to reach a contradiction, there is a sequence $(p_i, h_i, \tilde{H}_{1,i}, \tilde{H}_{2,i})$ so that

$$(F_i, \lambda_i) = \Big( p_i - \tilde{h}_i + \tilde{H}_{1,i} + \tilde{H}_{2,i}, \|\tilde{h}_i\|_1 - \|\tilde{H}_{1,i} + \tilde{H}_{2,i}\|_1 \Big)$$

has $F_i$ converges to $F_{\lim}$ with respect to $L_1$-norm and $\lambda_i$ converges to $\lambda_{\lim}$. We claim then that $(F_{\lim}, \lambda_{\lim}) \in \mathcal{S}_+$. First, notice that $\|p_i\|_1 \leq \|F_i\|_1 + \|\tilde{h}_i\|_1 + \|\tilde{H}_{1,i}\|_1 + \|\tilde{H}_{2,i}\|_1 \leq \|F_i\|_1 + 3$. Therefore, there must be a $k$ such that for any $i \geq k$, $\|p_i\|_1 \leq \|F_{\lim}\|_1 + 4$.

Then, since an $L_1$ ball in $\mathcal{P}_n$ with radius $\|F_{\lim}\|_1 + 4$ is compact, there must be a subsequence of $(p_i, h_i, \tilde{H}_{1,i}, \tilde{H}_{2,i})$ such that $p_i$ converges to $p_{\lim}$ for some $p_{\lim} \in \mathcal{P}_n$. Notice that due to the positivity of $\tilde{h}_i$ and $\tilde{H}_{2,i}$, it must be $\tilde{h}_i = (F_i - p_i)^-\mathbb{1}(z \geq c)$, $\tilde{H}_{2,i} = (F_i - p_i)^+\mathbb{1}(z \geq c)$ and $\tilde{H}_{1,i} = (F_i - p_i)\mathbb{1}(z < c)$. Therefore, $\tilde{h}_i, \tilde{H}_{1,i}$ and $\tilde{H}_{2,i}$ converge to $\tilde{h}_{\lim} = (F_{\lim} - p_{\lim})^-\mathbb{1}(z \geq c)$, $\tilde{H}_{2,\lim} = (F_{\lim} - p_{\lim})^+\mathbb{1}(z \geq c)$ and $\tilde{H}_{1,\lim} = (F_{\lim} - p_{\lim})\mathbb{1}(z < c)$ respectively. Then by taking $(p_{\lim}, h_{\lim}, \tilde{H}_{1,\lim}, \tilde{H}_{2,\lim})$, one can see that $(F_{\lim}, \lambda_{\lim}) \in \mathcal{S}_+$.

It only remains to verify that if there exists a polynomial $p$ of degree $n$ such that

$$\mathop{\mathbf{E}}_{z \sim \mathcal{N}_1}[|p(z)|\mathbb{1}(z \leq c)] < \mathop{\mathbf{E}}_{z \sim \mathcal{N}_1}[p(z)\mathbb{1}(z \geq c)] \ ,$$

then $(0, 1) \in \mathcal{S}_+$. Let $p$ be the polynomial satisfying the above. We can simply take $\tilde{h}_i = (p_i)^+\mathbb{1}(z \geq c)$, $\tilde{H}_{2,i} = (p_i)^-\mathbb{1}(z \geq c)$ and $\tilde{H}_{1,i} = -p_i\mathbb{1}(z < c)$. This gives

$$\Big( p_i - \tilde{h}_i + \tilde{H}_{1,i} + \tilde{H}_{2,i}, \|\tilde{h}_i\|_1 - \|\tilde{H}_{1,i} + \tilde{H}_{2,i}\|_1 \Big) = (0, k) \in \mathcal{S}_+ \ ,$$

where

$$k = \|(p_i)^+\mathbb{1}(z \geq c)\|_1 - \|(p_i)^-\mathbb{1}(z \geq c) - p_i\mathbb{1}(z < c)\|_1$$
$$= \|(p_i)^+\mathbb{1}(z \geq c)\|_1 - \|(p_i)^-\mathbb{1}(z \geq c)\|_1 - \|p_i\mathbb{1}(z < c)\|_1$$
$$= \mathop{\mathbf{E}}_{z \sim \mathcal{N}_1}[p(z)\mathbb{1}(z \geq c)] - \mathop{\mathbf{E}}_{z \sim \mathcal{N}_1}[|p(z)|\mathbb{1}(z \leq c)] > 0$$

Then, by properly rescaling $(0, k)$, we get $(0, 1) \in \mathcal{S}_+$. This completes the proof. $\qquad\square$

Given Claim C.2, to prove Lemma 3.3, it only remains to show that there does not exist a polynomial $p$ of degree at most $n$ such that

$$\mathop{\mathbf{E}}_{z \sim \mathcal{N}_1}[|p(z)|\mathbb{1}(z \leq c)] < \mathop{\mathbf{E}}_{z \sim \mathcal{N}_1}[p(z)\mathbb{1}(z \geq c)] .$$

First, the above condition implies that $\mathbf{E}_{z \sim \mathcal{N}_1}[|p(z)|] < 2\,\mathbf{E}_{z \sim \mathcal{N}_1}[|p(z)|\mathbb{1}(z \geq c)]$. Using the Cauchy–Schwarz inequality, we get

$$\mathop{\mathbf{E}}_{z \sim \mathcal{N}_1}[|p(z)|] < 2 \mathop{\mathbf{E}}_{z \sim \mathcal{N}_1}[|p(z)|^2]^{1/2} \left(\mathop{\mathbf{Pr}}_{z \sim \mathcal{N}_1}[z \geq c]\right)^{1/2} . \tag{3}$$

We then give the following claim.

**Claim C.4.** *Let $p : \mathbb{R} \mapsto \mathbb{R}$ be any polynomial of degree at most $n$. Then, it holds that*

$$\mathop{\mathbf{E}}_{z \sim \mathcal{N}_1}[|p(z)|^2]^{1/2} \leq 3^n \mathop{\mathbf{E}}_{z \sim \mathcal{N}_1}[|p(z)|] .$$

*Proof for Claim C.4.* The proof is based on the following well-known fact: for any function $g : \mathbb{R} \mapsto \mathbb{R}$, it holds that (from Holder's inequality)

$$\mathop{\mathbf{E}}_{z \sim \mathcal{N}_1}[|g(z)|^2] = \mathop{\mathbf{E}}_{z \sim \mathcal{N}_1}[|g(z)|^{2/3}|g(z)|^{4/3}] \leq \mathop{\mathbf{E}}_{z \sim \mathcal{N}_1}[|g(z)|]^{2/3} \mathop{\mathbf{E}}_{z \sim \mathcal{N}_1}[|g(z)|^4]^{1/3} ,$$

thus

$$\mathop{\mathbf{E}}_{z \sim \mathcal{N}_1}[|g(z)|^2]^{1/2} \leq \mathop{\mathbf{E}}_{z \sim \mathcal{N}_1}[|g(z)|]^{1/3} \mathop{\mathbf{E}}_{z \sim \mathcal{N}_1}[|g(z)|^4]^{1/6} . \tag{4}$$

We then use Fact C.5 (Gaussian Hypercontractivity) to bound the term $\mathbf{E}_{z \sim \mathcal{N}_1}[|g(z)|^4]^{1/6}$.

**Fact C.5** (Gaussian Hypercontractivity)**.** *Let $p : \mathbb{R} \mapsto \mathbb{R}$ be any polynomial of degree at most $l$. Then, for $q \geq 2$, it holds*

$$\mathop{\mathbf{E}}_{x \sim \mathcal{N}_1}[|p(x)|^q] \leq (q-1)^{ql/2} \left(\mathop{\mathbf{E}}_{x \sim \mathcal{N}_1}[p^2(x)]\right)^{q/2} .$$

Now we apply Fact C.5 and choose $l = n$ and $q = 4$. This implies $\mathbf{E}_{z \sim \mathcal{N}_1}[|p(z)|^4] \leq 3^{2n}\,\mathbf{E}_{z \sim \mathcal{N}_1}[|p(z)|^2]^2$, which is $\mathbf{E}_{z \sim \mathcal{N}_1}[|p(z)|^4]^{1/6} \leq 3^{n/3}\,\mathbf{E}_{z \sim \mathcal{N}_1}[|p(z)|^2]^{1/3}$. Plugging it into Equation (4), we get that for any polynomial $p$ of at most degree-$n$,

$$\mathop{\mathbf{E}}_{z \sim \mathcal{N}_1}[|p(z)|^2]^{1/2} \leq 3^{n/3} \mathop{\mathbf{E}}_{z \sim \mathcal{N}_1}[|p(z)|]^{1/3} \mathop{\mathbf{E}}_{z \sim \mathcal{N}_1}[|p(z)|^2]^{1/3} .$$

This implies

$$\mathop{\mathbf{E}}_{z \sim \mathcal{N}_1}[|p(z)|^2]^{1/2} \leq 3^n \mathop{\mathbf{E}}_{z \sim \mathcal{N}_1}[|p(z)|] .$$

$\qquad\square$

Applying Claim C.4 to Equation (3), we get

$$\mathop{\mathbf{E}}_{z \sim \mathcal{N}_1}[|p(z)|] < 2 \cdot 3^n \mathop{\mathbf{E}}_{z \sim \mathcal{N}_1}[|p(z)|] \left(\mathop{\mathbf{Pr}}_{z \sim \mathcal{N}_1}[z \geq c]\right)^{1/2} .$$

From our choice of the parameter $c$, we have that $\mathbf{Pr}_{z \sim \mathcal{N}_1}[z \geq c] \leq 3^{-2n}/4$, thus

$$\mathop{\mathbf{E}}_{z \sim \mathcal{N}_1}[|p(z)|] < \mathop{\mathbf{E}}_{z \sim \mathcal{N}_1}[|p(z)|] ,$$

contradiction. Therefore, such a polynomial $p$ cannot exist. Thus, a function $g$ satisfying the requirements in the body of the lemma exists.

## C.2  Proof of Lemma 3.4

We let $D$ be the unique distribution such that $z \sim \mathcal{N}_1$ and $\mathbf{E}_{(z,y)\sim D}[y|z=z'] = g(z')$ where $g$ is the function from Lemma 3.3. We claim that $D$ satisfies the conditions in this lemma's statement. Condition (i) is immediate from the definition of $D$. Then Condition (ii) is immediately implied from Condition (a) in Lemma 3.3.

For Condition (iii), Condition (b) in Lemma 3.3 implies $\mathbf{E}_{z\sim\mathcal{N}_1}[g(z)] = 0$ which immediately implies $\mathbf{E}_{(z,y)\sim D}[y] = 0$. To show that

$$\mathop{\mathbf{E}}_{(z,y)\sim D}[z^k] = \mathop{\mathbf{E}}_{(z,y)\sim D}[z^k \mid y = 1] = \mathop{\mathbf{E}}_{(z,y)\sim D}[z^k \mid y = -1]$$

for all $k \in [n]$, notice that for all $k \in [n]$,

$$\mathop{\mathbf{E}}_{(z,y)\sim D}[z^k] = \mathop{\mathbf{Pr}}_{(z,y)\sim D}[y = 1] \mathop{\mathbf{E}}_{(z,y)\sim D}[z^k \mid y = 1] + \mathop{\mathbf{Pr}}_{(z,y)\sim D}[y = -1] \mathop{\mathbf{E}}_{(z,y)\sim D}[z^k \mid y = -1] . \quad (5)$$

Moreover, from Condition (ii) of Lemma 3.3, we have

$$\mathop{\mathbf{E}}_{(z,y)\sim D}[g(z)z^k] = \mathop{\mathbf{E}}_{(z,y)\sim D}[yz^k]$$

$$= \mathop{\mathbf{Pr}}_{(z,y)\sim D}[y = 1] \mathop{\mathbf{E}}_{(z,y)\sim D}[z^k \mid y = 1] - \mathop{\mathbf{Pr}}_{(z,y)\sim D}[y = -1] \mathop{\mathbf{E}}_{(z,y)\sim D}[z^k \mid y = -1]$$

$$= 0 . \quad (6)$$

Taking the difference between Equation (5) and Equation (6) gives

$$\mathop{\mathbf{E}}_{(z,y)\sim D}[z^k] = 2 \mathop{\mathbf{Pr}}_{(z,y)\sim D}[y = -1] \mathop{\mathbf{E}}_{(z,y)\sim D}[z^k \mid y = -1] .$$

Plugging in that $\mathbf{Pr}_{(z,y)\sim D}[y = -1] = 1/2$, which follows from $\mathbf{E}_{(z,y)\sim D}[y] = 0$ and $y$ is supported on $\{\pm 1\}$, we get

$$\mathop{\mathbf{E}}_{(z,y)\sim D}[z^k] = \mathop{\mathbf{E}}_{(z,y)\sim D}[z^k \mid y = -1] .$$

Then, we have

$$\mathop{\mathbf{E}}_{(z,y)\sim D}[z^k] = \mathop{\mathbf{E}}_{(z,y)\sim D}[z^k \mid y = 1] \mathop{\mathbf{Pr}}_{(z,y)\sim D}[y = 1] + \mathop{\mathbf{E}}_{(z,y)\sim D}[z^k \mid y = -1] \mathop{\mathbf{Pr}}_{(z,y)\sim D}[y = -1] ;$$

therefore,

$$\mathop{\mathbf{E}}_{(z,y)\sim D}[z^k] = \mathop{\mathbf{E}}_{(z,y)\sim D}[z^k \mid y = -1] .$$

Thus, Condition (iii) holds.

Now, we show the fourth condition. We show the following claim.

**Claim C.6.** *It holds that $\chi^2(D_+, \mathcal{N}_1) = O(1)$ .*

*Proof.*

$$\chi^2(D_+, \mathcal{N}_1) = \int_{\mathbb{R}} \frac{P_{D_+}(z)^2}{P_{\mathcal{N}_1}(z)} dz - 1$$

$$= \int_{\mathbb{R}} P_{D_+}(z) \frac{P_{D_+}(z)}{P_{\mathcal{N}_1}(z)} dz - 1$$

$$\leq \int_{\mathbb{R}} P_{D_+}(z) / \mathop{\mathbf{Pr}}_{(z,y)\sim D_f}[y = 1] dz - 1$$

$$= \mathop{\mathbf{Pr}}_{(z,y)\sim D}[y = 1]^{-1} \int_{\mathbb{R}} P_{D_+}(z) dz - 1$$

$$= \mathop{\mathbf{Pr}}_{(z,y)\sim D}[y = 1]^{-1} - 1 = O(1) ,$$

where the inequality follows from the fact that

$$P_{\mathcal{N}_1}(z) = P_{D_+}(z) \mathop{\mathbf{Pr}}_{(z,y)\sim D}[y = 1] + P_{D_-}(z) \mathop{\mathbf{Pr}}_{(z,y)\sim D}[y = -1] \geq P_{D_+}(z) \mathop{\mathbf{Pr}}_{(z,y)\sim D}[y = 1] ,$$

which implies $P_{D_+}(z)/P_{\mathcal{N}_1}(z) \leq 1/\mathbf{Pr}_{(z,y)\sim D}[y = 1]$. The same holds for $\chi^2(D_-, \mathcal{N}_1)$. This completes the proof of Claim C.6. □

The proof of $\chi^2(D_-, \mathcal{N}_1) = O(1)$ is similar to Claim C.6. This completes the proof of Lemma 3.4.

## C.3   Construction of the Alternative Hypothesis Distribution Set $\mathcal{D}$

**Lemma C.7.** *For any sufficiently small $\alpha > 0$, there exists a distribution family $\mathcal{D} = \{D_{\mathbf{v}} : \mathbf{v} \in V\}$ supported on $\mathbb{R}^d \times \{\pm 1\}$ where $|\mathcal{D}| = 2^{d^{\Omega(1)}}$ satisfying the following:*

*(i) For any $D_{\mathbf{v}} \in \mathcal{D}$ and $(\mathbf{x}, y) \sim D_{\mathbf{v}}$, the marginal $D_{\mathbf{x}} = \mathcal{N}_d$;*

*(ii) For any $D_{\mathbf{v}} \in \mathcal{D}$ and $(\mathbf{x}, y) \sim D_{\mathbf{v}}$, $\mathbf{E}_{(\mathbf{x},y) \sim D_{\mathbf{v}}}[y | \mathbf{x} = \mathbf{x}'] = 1$ for all $\mathbf{x}' \in \mathbb{R}^d$ such that $\langle \mathbf{v}, \mathbf{x}' \rangle \geq \Phi^{-1}(1 - \alpha)$;*

*(iii) Let $D_{\mathrm{null}}$ be the joint distribution of $(\mathbf{x}, y)$ such that $\mathbf{x} \sim \mathcal{N}_d$ and $y = 1$ with probability $1/2$ independent of $\mathbf{x}$. Then for any $D_{\mathbf{u}}, D_{\mathbf{v}} \in \mathcal{D}$, $\chi_{D_{\mathrm{null}}}(D_{\mathbf{u}}, D_{\mathbf{v}}) = O(1)$ if $\mathbf{u} = \mathbf{v}$ and $\chi_{D_{\mathrm{null}}}(D_{\mathbf{u}}, D_{\mathbf{v}}) = d^{-\Omega(\log 1/\alpha)}$ if $\mathbf{u} \neq \mathbf{v}$.*

Here, we give two lemmas from [DKPZ21] to construct a large set of distributions on $\mathbb{R}^d \times \{\pm 1\}$ with small correlations. For a matrix $\mathbf{A} \in \mathbb{R}^{m \times n}$, we use $\|\mathbf{A}\|_2$ to denote the spectral norm.

**Lemma C.8** (Correlation Lemma: Lemma 2.3 from [DKPZ21]). *Let $g : \mathbb{R}^m \mapsto \mathbb{R}$ and $\mathbf{U}, \mathbf{V} \in \mathbb{R}^{m \times d}$ with $m \leq d$ be linear maps such that $\mathbf{U}\mathbf{U}^T = \mathbf{V}\mathbf{V}^T = \mathbf{I}$ where $\mathbf{I}$ is the $m \times m$ identity matrix. Then, we have that*

$$\mathop{\mathbf{E}}_{\mathbf{x} \sim \mathcal{N}_d}[g(\mathbf{U}\mathbf{x})g(\mathbf{V}\mathbf{x})] \leq \sum_{t=0}^{\infty} \|\mathbf{U}\mathbf{V}^T\|_2^t \mathop{\mathbf{E}}_{\mathbf{x} \sim \mathcal{N}_m}[(g^{[t]}(\mathbf{x}))^2] \,,$$

*where $g^{[t]}$ denote the degree-$t$ Hermite part of $g$.*

Note that we will use Lemma C.8 with $m = 1$ and $n = d$ so that the linear maps $\mathbf{U}$ and $\mathbf{V}$ become vectors. We then introduce the following well-known fact, which states that there are exponentially many near-orthogonal vectors in $\mathbb{R}^d$.

**Fact C.9** (Near-orthogonal Vectors: Fact 2.6 from [DKPZ21]). *For any $0 < c < 1/2$, there exists a set $V \subseteq \mathbb{S}^{d-1}$ such that $|V| = 2^{d^{\Omega(c)}}$ and for any $\mathbf{u}, \mathbf{v} \in V$, $\langle \mathbf{u}, \mathbf{v} \rangle \leq d^{c-1/2}$ .*

*Proof of Lemma C.7.* We define $\mathcal{D}$ in the following manner. We let $D'$ be the distribution on $\mathbb{R} \times \{\pm 1\}$ in Lemma 3.4 for $n = c \log(1/\alpha)$ where $c$ is a sufficiently small universal constant such that $3^{-2n}/4 \geq \alpha$. We take $V$ to be the set of vectors in Fact C.9 with $c = 1/4$. Then we consider the set of distributions $\mathcal{D} = \{D_{\mathbf{v}} : \mathbf{v} \in V\}$ where $D_{\mathbf{v}}$ is the unique distribution on $\mathbb{R}^d \times \{\pm 1\}$ generated in the following way. We first sample $(z, y) \sim D'$ and $\mathbf{x}_{\perp} \sim \mathcal{N}_{d-1}$. Then let $\mathbf{B}_{\perp} \in \mathbb{R}^{d \times (d-1)}$ be a matrix whose columns form an (arbitrary) orthonormal basis for the orthogonal complement of $\mathbf{v}$. Let $\mathbf{x} = \mathbf{B}_{\perp}\mathbf{x}_{\perp} + z\mathbf{v}$ and we define the distribution $D_{\mathbf{v}} = D_{(\mathbf{x},y)}$. This effectively embeds the distribution $D'$ along the hidden direction $\mathbf{v}$ in $D_{\mathbf{v}}$.

To establish Condition (i) in this lemma, the definition of $\mathbf{x}$ combined with the fact that $D_z = \mathcal{N}_1$ (from Condition (i) in Lemma 3.4) implies the marginal distribution $D_{\mathbf{x}} = \mathcal{N}_d$.

Then, for Condition (ii) in this lemma, we have that for any $\mathbf{x}' \in \mathbb{R}^d$ such that $\langle \mathbf{v}, \mathbf{x}' \rangle \geq \Phi^{-1}(1 - \alpha)$,

$$\mathop{\mathbf{E}}_{(\mathbf{x},y) \sim D_{\mathbf{v}}}[y | \mathbf{x} = \mathbf{x}'] = \mathop{\mathbf{E}}_{(z,y) \sim D'}[y = 1 | z = \langle \mathbf{v}, \mathbf{x}' \rangle] \,,$$

from the definition of $D_{\mathbf{v}}$. Since $3^{-2n}/4 \geq \alpha$, we have $\langle \mathbf{v}, \mathbf{x}' \rangle \geq \Phi^{-1}(1 - \alpha) \geq \Phi^{-1}(1 - 3^{-2n}/4)$. Therefore, using the second condition in Lemma 3.4, we get

$$\mathop{\mathbf{E}}_{(\mathbf{x},y) \sim D_{\mathbf{v}}}[y | \mathbf{x} = \mathbf{x}'] = \mathop{\mathbf{E}}_{(z,y) \sim D'}[y | z = \langle \mathbf{v}, \mathbf{x}' \rangle] = 1 \,.$$

For Condition (iii) in this lemma, notice that Condition (iii) in Lemma 3.4 implies

$$\mathop{\mathbf{Pr}}_{(\mathbf{x},y) \sim D_{\mathbf{v}}}[y = 1] = \mathop{\mathbf{Pr}}_{(\mathbf{x},y) \sim D_{\mathbf{v}}}[y = -1] = 1/2 \,.$$

Then,

$$\chi_{D_{\mathrm{null}}}(D_{\mathbf{v}}, D_{\mathbf{v}}) = \mathop{\mathbf{Pr}}_{(\mathbf{x},y) \sim D_{\mathbf{v}}}[y = 1]\chi^2(D_{\mathbf{v}}^+, \mathcal{N}_d) + \mathop{\mathbf{Pr}}_{(\mathbf{x},y) \sim D_{\mathbf{v}}}[y = -1]\chi^2(D_{\mathbf{v}}^-, \mathcal{N}_d)$$

$$= \frac{1}{2}\chi^2(D_{\mathbf{v}}^+, \mathcal{N}_d) + \frac{1}{2}\chi^2(D_{\mathbf{v}}^-, \mathcal{N}_d) \,.$$

We show that the first term $\chi^2(D_{\mathbf{v}}^+, \mathcal{N}_d) = O(1)$. $\chi^2(D_{\mathbf{v}}^-, \mathcal{N}_d) = O(1)$ can be shown in the exact same way. Notice that

$$
\begin{aligned}
\chi^2(D_{\mathbf{v}}^+, \mathcal{N}_d) &= \int_{\mathbb{R}^d} \frac{P_{D_{\mathbf{v}}^+}(\mathbf{u})^2}{P_{\mathcal{N}_d}(\mathbf{u})} d\mathbf{u} - 1 \\
&= \int_{\mathbb{R}} \int_{\mathbb{R}^{d-1}} \frac{P_{D_{\mathbf{v}}^+}(\mathbf{B}_\perp \mathbf{u} + z\mathbf{v})^2}{P_{\mathcal{N}_d}(\mathbf{B}_\perp \mathbf{u} + z\mathbf{v})} d\mathbf{u} dz - 1 \\
&= \int_{\mathbb{R}} \int_{\mathbb{R}^{d-1}} \frac{P_{\mathcal{N}_{d-1}}(\mathbf{u})^2 P_{D'^+}(z)^2}{P_{\mathcal{N}_{d-1}}(\mathbf{u}) P_{\mathcal{N}_1}(z)} d\mathbf{u} dz - 1 \\
&= \int_{\mathbb{R}^{d-1}} P_{\mathcal{N}_{d-1}}(\mathbf{u}) d\mathbf{u} \int_{\mathbb{R}} \frac{P_{D'^+}(z)^2}{P_{\mathcal{N}_1}(z)} dz - 1 \\
&= \chi^2(D'^+, \mathcal{N}_1) = O(1) ,
\end{aligned}
$$

where the last equality follows from Condition (iv) of Lemma 3.4. Therefore

$$
\chi_{D_{\mathrm{null}}}(D_{\mathbf{v}}, D_{\mathbf{v}}) = \frac{1}{2}\chi^2(D_{\mathbf{v}}^+, \mathcal{N}_d) + \frac{1}{2}\chi^2(D_{\mathbf{v}}^-, \mathcal{N}_d) = O(1) .
$$

For $D_{\mathbf{u}}, D_{\mathbf{v}} \in \mathcal{D}$ where $\mathbf{u} \neq \mathbf{v}$, we have

$$
\begin{aligned}
\chi_{D_{\mathrm{null}}}(D_{\mathbf{u}}, D_{\mathbf{v}}) &= \mathbf{Pr}[y = 1]\chi_{\mathcal{N}_d}(D_{\mathbf{u}}^+, D_{\mathbf{v}}^+) + \mathbf{Pr}[y = -1]\chi_{\mathcal{N}_d}(D_{\mathbf{u}}^-, D_{\mathbf{v}}^-) \\
&= \frac{1}{2}\chi_{\mathcal{N}_d}(D_{\mathbf{u}}^+, D_{\mathbf{v}}^+) + \frac{1}{2}\chi_{\mathcal{N}_d}(D_{\mathbf{u}}^-, D_{\mathbf{v}}^-) .
\end{aligned}
$$

We show that $\chi_{\mathcal{N}_d}(D_{\mathbf{u}}^+, D_{\mathbf{v}}^+) = d^{-\Omega(\log \frac{1}{\alpha})}$. $\chi_{\mathcal{N}_d}(D_{\mathbf{u}}^-, D_{\mathbf{v}}^-) = d^{-\Omega(\log \frac{1}{\alpha})}$ can be shown in the exact same way. Notice that

$$
\begin{aligned}
\chi_{\mathcal{N}_d}(D_{\mathbf{u}}^+, D_{\mathbf{v}}^+) &= \int_{\mathbb{R}^d} \frac{P_{D_{\mathbf{u}}^+}(\mathbf{w}) P_{D_{\mathbf{v}}^+}(\mathbf{w})}{P_{\mathcal{N}_d}(\mathbf{w})} d\mathbf{w} - 1 \\
&= \int_{\mathbb{R}^d} \frac{P_{D_{\mathbf{u}}^+}(\mathrm{proj}_{\perp \mathbf{u}}(\mathbf{w}) + \mathrm{proj}_{\mathbf{u}}(\mathbf{w})) P_{D_{\mathbf{v}}^+}(\mathrm{proj}_{\perp \mathbf{v}}(\mathbf{w}) + \mathrm{proj}_{\mathbf{v}}(\mathbf{w}))}{P_{\mathcal{N}_d}(\mathbf{w})} d\mathbf{w} - 1 \\
&= \int_{\mathbb{R}^d} \frac{P_{\mathcal{N}_{d-1}}(\mathbf{w}^{\perp \mathbf{u}}) P_{D'^+}(\mathbf{w}^{\mathbf{u}}) P_{\mathcal{N}_{d-1}}(\mathbf{w}^{\perp \mathbf{v}}) P_{D'^+}(\mathbf{w}^{\mathbf{v}})}{P_{\mathcal{N}_d}(\mathbf{w})} d\mathbf{w} - 1 \\
&= \int_{\mathbb{R}^d} P_{\mathcal{N}_d}(\mathbf{w}) \frac{P_{D'^+}(\mathbf{w}^{\mathbf{u}})}{P_{\mathcal{N}_1}(\mathbf{w}^{\mathbf{u}})} \frac{P_{D'^+}(\mathbf{w}^{\mathbf{v}})}{P_{\mathcal{N}_1}(\mathbf{w}^{\mathbf{v}})} d\mathbf{w} - 1 .
\end{aligned}
$$

Let $g : \mathbb{R} \to \mathbb{R}$ be $g(z) \stackrel{\text{def}}{=} P_{D'^+}(z)/P_{\mathcal{N}_1}(z)$ and apply Lemma C.8 and Condition (iii) of Lemma 3.4, we get

$$
\begin{aligned}
\chi_{\mathcal{N}_d}(D_{\mathbf{u}}^+, D_{\mathbf{v}}^+) &\leq \left(1 + \sum_{t=n+1}^{\infty} d^{-\Omega(t)} \mathop{\mathbf{E}}_{z \sim \mathcal{N}_1}\left[\left(g^{[t]}(z)\right)^2\right]\right) - 1 \\
&\leq d^{-\Omega(n)}\chi_{\mathcal{N}_1}^2(D'^+, \mathcal{N}_1) \\
&= d^{-\Omega(\log \frac{1}{\alpha})} ,
\end{aligned}
$$

where the last equality follows form Condition (iv) of Lemma 3.4. This completes the proof of Lemma Lemma C.7. $\qquad \square$

## C.4   Proof of Theorem C.1

Let $\mathcal{D}$ be the set of distributions in Lemma C.7. We also let $D_{\mathrm{null}}$ be the joint distribution of $(\mathbf{x}, y)$ such that $\mathbf{x} \sim \mathcal{N}_d$ and $y \sim \mathrm{Bern}(1/2)$ independent of $\mathbf{x}$. We consider the decision problem of $\mathcal{B}(\mathcal{D}, D_{\mathrm{null}})$. Using Lemma A.4 with $\gamma' = d^{-\log(\frac{1}{\alpha})}$, we can see that any SQ algorithm that solves $\mathcal{B}(\mathcal{D}, D_{\mathrm{null}})$ requires either queries of tolerance at most $d^{-\Omega(\log \frac{1}{\alpha})}$ or makes at least $2^{d^{\Omega(1)}}$ queries.

We then reduce the decision problem $\mathcal{B}(\mathcal{D}, D_{\mathrm{null}})$ to reliably learning $\alpha$-biased LTFs with $\epsilon < \alpha/3$ accuracy. Let $D$ be an instance of $\mathcal{B}(\mathcal{D}, D_{\mathrm{null}})$ which we need to decide either $D = D_{\mathrm{null}}$ or

$D \in \mathcal{D}$. Let $A$ be an algorithm that reliably learns $\alpha$-biased LTFs with $\epsilon < \alpha/3$ accuracy and succeeds with 2/3 probability. We can simply give the i.i.d. samples from the distribution $D$ to algorithm $A$. Suppose that $A$ succeeds and outputs a hypothesis function $h$. If $D = D_{\mathrm{null}}$, then since $A$ promises $R_+(h, D_{\mathrm{null}}) \leq \epsilon$. Since $D_{\mathrm{null}}$ has $y \sim \mathrm{Bern}(1/2)$ independent of $\mathbf{x}$, this implies $\mathbf{Pr}_{(\mathbf{x},y) \sim D}[h(\mathbf{x}) = 1] \leq \epsilon$.

Then we argue that if we are in the alternative case, i.e., $D \in \mathcal{D}$, then given that the algorithm succeeds, we must have $\mathbf{Pr}_{(\mathbf{x},y) \sim D}[h(\mathbf{x}) = 1] \geq \alpha - \epsilon$ instead. We assume $\mathbf{Pr}_{(\mathbf{x},y) \sim D}[h(\mathbf{x}) = 1] < \alpha - \epsilon$ and prove a contradiction. If $D = D_{\mathbf{v}} \in \mathcal{D}$, then by the first property in Lemma C.7, there must be an $\alpha$-biased LTF $c$ such that $\mathbf{Pr}_{(\mathbf{x},y) \sim D}[c(\mathbf{x}) = 1] \geq \alpha$ and $\mathbf{Pr}_{(\mathbf{x},y) \sim D}[c(\mathbf{x}) = 1 \wedge y = -1] = 0$. Combining this with the fact that $\mathbf{Pr}_{(\mathbf{x},y) \sim D}[y = 1] = 1/2$, we have $R_-(c; D) = \mathbf{Pr}[c(\mathbf{x}) = -1 \wedge y = 1] \leq 1/2 - \alpha$. Therefore, the output hypothesis $h$ of $A$ has to satisfy

$$R_-(h; D) \leq 1/2 - \alpha + \epsilon \,.$$

However, since $\mathbf{Pr}_{(\mathbf{x},y) \sim D}[h(\mathbf{x}) = 1] < \alpha - \epsilon$, we have

$$R_-(h; D) = \mathop{\mathbf{Pr}}_{(\mathbf{x},y) \sim D}[h(\mathbf{x}) \neq 1 \wedge y = 1] \geq \mathop{\mathbf{Pr}}_{(\mathbf{x},y) \sim D}[y = 1] - \mathop{\mathbf{Pr}}_{(\mathbf{x},y) \sim D}[h(\mathbf{x}) = 1] > 1/2 - \alpha + \epsilon$$

which contradicts the above. Thus, it has to be $\mathbf{Pr}_{(\mathbf{x},y) \sim D}[h(\mathbf{x}) = 1] > \alpha - \epsilon$.

If $D = D_{\mathrm{null}}$, then $\mathbf{Pr}_{(\mathbf{x},y) \sim D}[h(\mathbf{x}) = 1] \leq \epsilon$. Otherwise, we have $\mathbf{Pr}_{(\mathbf{x},y) \sim D}[h(\mathbf{x}) = 1] \geq \alpha - \epsilon$. Since the gap between them is at least a constant $\alpha - 2\epsilon \geq \alpha/3$ and $\mathbf{Pr}_{(\mathbf{x},y) \sim D}[h(\mathbf{x}) = 1]$ can be estimated to inverse exponential accuracy on samples with high probability. Thus, we can distinguish the two cases of $\mathcal{B}(\mathcal{D}, D_{\mathrm{null}})$ with high probability.

