# OpenReview forum: "Reliable Learning of Halfspaces under Gaussian Marginals"
_NeurIPS.cc/2024/Conference — NeurIPS 2024 spotlight_

### Official Review · Reviewer_ezCR · 2024-06-22

**Soundness:** 3
**Presentation:** 3
**Contribution:** 3
**Rating:** 7
**Confidence:** 2

**Summary:**

This paper studies the problem of agnostic reliable learning with Gaussian margin. It gives a novel algorithm with improved running time and sample complexity bound and this suggests that agnostic reliable learning is easier than agnostic learning. It also gives a Statistical Query lower bound matching some terms of the upper bound as an evidence that the upper bound is tight.

**Strengths:**

1. The result of this paper is novel and complete.

   a. The running time and sample complexity of the proposed algorithm is a big improvement from the previous $d^{O(1/\epsilon^2)}$ and the algorithm is completely new.

   b. The algorithm, lower bound and analysis are all highly non-trivial and technically sophisticated, showing a deep understanding of the problem. I think there's lots of technical novelties in the paper.

   c. The fact that there's a separation between agnostic reliable learning and agnostic learning is interesting and it's a contribution to formally establish this.

   d. The paper gives a non-trivial SQ lower bound matching some terms of the upper bound, suggesting its tightness.

2. The writing of the paper is decent. Enough background knowledge is given. Math related parts is clearly defined and the proof is rigorous. The paper is organized and non-technical part is not hard to follow.

**Weaknesses:**

1. There are some room for improvement in writing.

   a) As someone who is quite familiar with PAC and many classical learning problems but not familiar with reliable learning/learning Gaussian margin halfspaces, I find the introduction to the problem could be more clear, especially I think you can do a better job explaining the adversary's strategy and behavior, the "corrupt negative labels are free" part is worth more insights.

   b) I do find the paper really technical, but I think it is intrinsic. As someone who is not familiar with your algorithmic framework [DKK+22] and some technical parts (for example, the use of polynomials), I think more explanation could be helpful.

   c) Some parts of the writing could be more clear, like "with high probability" in the definition.

2. I do find the SQ lower bound a bit weak. It doesn't have a dependence on $\epsilon$, is there any known lower bounds on $\epsilon$?

3. The agnostic learning lower bound doesn't have a dependence on $\alpha$, but the agnostic reliable learning bound has such a dependence, why? You assumed that $\alpha$ is a constant, if it's not, is there any complications in your conclusion?

**Questions:**

1. In this paper and some other papers, the lower bounds are SQ lower bounds. As far as I know, SQ lower bounds are weaker than PAC lower bounds (SQ learnable implies PAC learnable but not the other way around), is there some difficulty to get a PAC lower bound for the problem?

2. As far as I know, active learning doesn't seem to help with agnostic learning halfspaces (Gaussian margin, arbitrary noise), could it improve the sample complexity bound for agnostic reliable learning (with Gaussian margin).

**Limitations:**

I think the authors could give some future directions and potential improvements.

---

> ### Author Rebuttal · Authors · 2024-08-07
>
> We thank the reviewer for the time and effort in reviewing our paper and for the positive feedback.
>
> > I do find the SQ lower bound a bit weak. It doesn't have a dependence on $\epsilon$, is there any known lower bounds on $\epsilon$?
>
> *Response:* The point of our SQ lower bound is that the joint dependence between the dimension $d$ and the bias $\alpha$ of the target concept is inherently quasi-polynomial, namely $d^{\Omega(\log(1/\alpha))}$ (assuming that $\alpha \geq 2\epsilon$). This lower bound holds for any $2\epsilon \leq\alpha$. Importantly, we provide the first such lower bound for reliable learning of Gaussian halfspaces. We do not know of a super-polynomial lower bound on $\epsilon$ and leave this as an open question for future investigation.
>
> At a more technical level, our construction establishes an SQ lower bound for a natural testing version of the problem (that is easier than learning). This testing problem is in fact solvable with SQ complexity $d^{\Theta(\log(1/\alpha))}$ (both upper and lower bound). The dependence on $\epsilon$ in our upper bound is due to the fact that we need to search for a good solution via a random walk, as explained in lines 108-130.
>
>
>
> > The agnostic learning lower bound doesn't have a dependence on $\alpha$, but the agnostic reliable learning bound has such a dependence, why? You assumed that $\alpha$ is a constant, if it's not, is there any complications in your conclusion?
>
> *Response:* For agnostically learning Gaussian halfspaces, the $d^{\mathrm{poly}(1/\epsilon)}$ SQ lower bound holds even for $\alpha=1/2$, which suggests that the agnostic problem does not become harder with smaller $\alpha$. We show in our work that this is not the case for the reliable learning task. Intuitively, in the agnostic model, the adversary can corrupt an $\epsilon$-fraction of the labels anywhere on the entire domain. On the other hand, in the reliable setting, the adversary can only corrupt labels inside the region $c(\mathbf{x})=-1$ (where $c$ is the target concept). As a result, one needs degree-$\mathrm{poly}(1/\epsilon)$ polynomials to not be fooled by noise in agnostic learning. While in reliable learning, our Lemma 2.3 shows that a degree-$O(\log (1/\alpha))$ polynomial suffices (where $\alpha$ is the bias of concept $c$ and $\alpha\geq \epsilon$). It is also worth mentioning that for the special case of $\alpha=1/2$, we get $\mathrm{poly}(d/\epsilon)$ runtime for reliable learning via an easy algorithm (see Lemma B.3).
>
> Finally, we remark that we do not assume anywhere that the parameter $\alpha$ is a constant. Our algorithm works for any value of $\alpha$ (even if it is not known a priori to the algorithm).
>
>
> > In this paper and some other papers, the lower bounds are SQ lower bounds. As far as I know, SQ lower bounds are weaker than PAC lower bounds (SQ learnable implies PAC learnable but not the other way around), is there some difficulty to get a PAC lower bound for the problem?
>
> *Response:* While SQ lower bounds are viewed as strong evidence of hardness in learning theory, we acknowledge that the SQ model is a restricted model of computation. So, in principle, it may possible to surpass SQ algorithms. On the other hand, there are very few such examples in the literature (and none for learning halfspaces with noise). Alternatively, one could try to prove reduction-based hardness (from a plausibly average-case hard problem, e.g., in cryptography). There are a few such reductions in the literature, and in some cases they have been directly inspired from the SQ-hard instances. A recent example that is relevant for us is for the task of agnostically learning Gaussian halfspaces, where [DKR23] (building on the SQ lower bounds of [DKZ20]) gives a cryptographic lower bound (assuming the subexponential hardness of LWE). Their technique requires a hard distribution whose marginal in the hidden direction is (roughly speaking) a mixture of periodic signals, which cannot be adapted for our hard distribution here. It remains an interesting open problem to establish reduction-based hardness for our problem.
>
> > As far as I know, active learning doesn't seem to help with agnostic learning halfspaces (Gaussian margin, arbitrary noise), could it improve the sample complexity bound for agnostic reliable learning (with Gaussian margin).
>
> *Response:* We agree with the reviewer that active learning does not seem to help with agnostic learning halfspaces under Gaussian marginals. It is not clear to us whether active learning can help in the reliable learning case. This is an interesting open question for future work.
>
> > I think the authors could give some future directions and potential improvements.
>
> *Response:* We are happy to add some future directions in the revised version. An immediate open question is whether one can improve the dependence of the runtime as a function of $\epsilon$ from quasi-polynomial down to polynomial.
>
> > the "corrupt negative labels are free" part is worth more insights.
>
> *Response:* In fact, corrupting negative labels is not free, as the adversary has an $\epsilon$ budget to corrupt negative labels. However, the adversary is restricted to only corrupting the "true" negative labeled samples as compared to the agnostic model.
>
> > More explanation on the use of polynomial
>
> *Response:* In lines 116-123, we explained the high-level idea of using polynomials to learn. We are happy to add further explanations to make the intution more transparent.
>
> [DKZ20] I. Diakonikolas, D. Kane, and N. Zarifis. Near-Optimal SQ Lower Bounds for Agnostically Learning Halfspaces and ReLUs under Gaussian Marginals. In Advances in Neural Information Processing Systems, NeurIPS, 2020.
>
> [DKR23] I. Diakonikolas, D. M. Kane, and L. Ren. Near-Optimal Cryptographic Hardness of Agnostically Learning Halfspaces and ReLU Regression under Gaussian Marginals. In International Conference on Machine Learning, ICML, 2023.

---

> > ### Comment · Reviewer_ezCR · 2024-08-09
> >
> > Thanks for your response! Actually the "corrupt negative labels are free" sentence is taken verbatim from your paper and you may want to change it.

---

### Official Review · Reviewer_ae3s · 2024-07-13

**Soundness:** 3
**Presentation:** 3
**Contribution:** 3
**Rating:** 7
**Confidence:** 2

**Summary:**

This paper considers learning halfspaces with Gaussian marginals in a reliable learning setting, where the learner has to guarantee that the error of the output classifier is less than $\epsilon$ (and we assume such a classifier exists). It is known that the reliable learning problem can be efficiently reduced to agnostic learning, but the sample complexity for agnostic learning is high ($d^{O(1/\epsilon)}$). This paper proposes an algorithm that has a much better sample complexity (polynomial in d, but still quasi-polynomial in $1/\epsilon$), and it provides a lower bound showing that one cannot do much better w.r.t. d.

**Strengths:**

- This paper considers an interesting learning theory problem.
- The results look nontrivial.
- The methods look sound, though I'm not familiar with the techniques used here and did not check the proofs.
- It is written clearly, and explains the intuition well.

**Weaknesses:**

- Though this is an interesting theory problem, it is not entirely clear to me how significant the results are. Specifically, what is the importance or implication of the computational separation between agnostic and reliable learning found in this paper, especially given that the proposed algorithm is still super-polynomial.

**Questions:**

N/A

---

> ### Author Rebuttal · Authors · 2024-08-07
>
> We thank the reviewer for the time and effort in reviewing our paper and for the positive feedback. We respond to the point raised by the reviewer regarding the significance of our result.
>
> Our result is the first to establish a computational separation between reliable and agnostic learning in the distribution-specific setting for the class of halfspaces. While our algorithm has superpolynomial runtime (as a function of the excess error $\epsilon$ and the bias $\alpha$), it achieves an exponential improvement over the previous best-known algorithm for the problem. Moreover, the underlying algorithmic technique is likely to be useful for related reliable learning tasks. It it worth noting that our algorithm runs in $\mathrm{poly}(d)$ time for any constant $\alpha$ and moderately small values of the parameter $\epsilon$ (namely up to $1/\mathrm{polylog}(d)$). On the other hand, for agnostic learning, one incurs super-polynomial time for *any* $\epsilon= o(1)$ (even if $\alpha=1/2$). Finally, we believe that our work provides fundamental insights into the task of reliable learning and lays the foundations for the development of practical algorithms for the task.

---

> > ### Comment · Reviewer_ae3s · 2024-08-12
> >
> > Thanks for the response. I will keep my score and support its acceptance.

---

### Official Review · Reviewer_ZrmL · 2024-07-15

**Soundness:** 4
**Presentation:** 3
**Contribution:** 3
**Rating:** 8
**Confidence:** 2

**Summary:**

This work studies agnostic learning of halfspaces in the reliable learning model, which guarantees a halfspace with nearly no false positives and a nearly optimal false negative rate, where the optimal false negative rate is defined relative to a class of halfspaces with no false positives. The authors prove sample and computational bounds for this learning task under a standard Gaussian distributional assumption, dramatically improving the previously known bounds that followed from reduction to general agnostic learning under the same distributional assumption. They also show a statistical query lower bound of $d^{\Omega(\log(1/\alpha))}$.

**Strengths:**

This work furthers our understanding of the sample and computational complexity of learning halfspaces under challenging noise models. The techniques used to obtain the algorithmic result are very interesting, and while technically involved, the overview of the proof approach in the main body is well-structured and modular (if still hard to follow as someone with little familiarity with the related work).

**Weaknesses:**

While the page-limits are restrictive, I would have benefited from some additional handholding even in the overview. The introduction of Lemma 2.5 was particularly opaque, for instance.


Typos/suggested edits:

Line 14. “The problem of learning halfspaces is one the classical”

Line 40. “has since been extensively studies”

Line 53. “minimizing a lost function”

Line 69. “as a reliable agnostic learning for Gaussian halfspaces”

Line 84. missing close parens

line 94. “reduce the fully reliable learning”

line 105. “This implies that that”

line 214. “Let D be joint distribution”

Algorithm 1 caption “General Halspaces”

Line 511/525 “Reliable learning halfspaces”

**Questions:**

In line 222, I’m confused about the signs. Doesn’t the interval $[t^*, \infty]$ correspond to positive labels, and don’t we want the expectation of p within this region to be negative?

How does the equality in line 241 follow from Lemma 2.5?

**Limitations:**

Yes, the authors address the limitations of the work.

---

> ### Author Rebuttal · Authors · 2024-08-07
>
> We thank the reviewer for the time and effort in reviewing our paper and for the positive feedback. Below, we provide specific responses to the points and questions raised by the reviewer.
>
> > In line 222, I’m confused about the signs. Doesn’t the interval $[t^*,\infty]$ correspond to positive labels, and don’t we want the expectation of $p$ within this region to be negative?
>
> *Response:* In Lemma 2.3, our goal is to show that $\mathbf{E}_D[y\,p(\mathbf{x})]$ is large. Therefore, ideally, we want the polynomial $p$ to be positive on regions with positive labels and negative on regions with negative labels.
>
> > How does the equality in line 241 follow from Lemma 2.5?
>
> *Response:* There is a typo in the equality in line 241, where there should be an $\Omega(\cdot)$ in the right-hand side. This follows from the fact that the correlation between two random $d$-dimensional unit vectors is at least $\Omega(1/\sqrt{d})$ with high probability. Lemma 2.5 gives a lower bound on $||\mathrm{proj}_V(\mathbf{v}^*)||_2$, which implies $|\langle \mathbf{v},\mathbf{v}^*\rangle |$ is sufficiently large for a randomly chosen unit vector $\mathbf{v}$.

---

> > ### Comment · Reviewer_ZrmL · 2024-08-12
> >
> > Thank you to the authors for answering my questions, this makes sense!

---

### Official Review · Reviewer_wFKn · 2024-07-17

**Soundness:** 4
**Presentation:** 4
**Contribution:** 3
**Rating:** 7
**Confidence:** 3

**Summary:**

This paper studies reliable learning of halfspaces in $d$ dimensions. Reliable learning is a framework in learning theory in which the learning algorithm is required to output a classifier $f$ satisfying:
- the probability that f makes a false-positive error is at most $\epsilon$
- The probability that f makes a false-negative error is at most $opt+ \epsilon$, where $opt$ is the smallest error rate achievable by a classifier in the class $\mathcal{G}$ that has zero false-positive error.

The work gives a reliable learning algorithm for the class of $\alpha$-biased halfspaces when the data marginal is the standard Gaussian distribution. A halfspace is $\alpha$-biased if on a Gaussian input it has probability at least $\alpha$ to take either of the two possible output values. The run-time of the algorithm is $d^{O(\log(min(1/\alpha,1/\epsilon)))}min(2^{\log(1/\epsilon)^{O(log(1/\alpha))}}
, 2^{poly(1/\epsilon)})$.

The algorithm first finds a candidate direction by estimating the Chow tensor. Consequently, the algorithm improves this hypothesis by performing a certain random walk.

It is shown that any statistical-query algorithm has to take at least $d^{\log 1/\alpha}$ time for this task. Statistical-query algorithm are a wide family of algorithms that includes virtually all algorithms studied in learning theory.

**Strengths:**

- Reliable learning is a natural framework asking for approximately-best classifier that makes false positive rarely. Considering halfspaces over Gaussian data is arguably the most natural setting to study. However, prior to this work little was known about this question.
- The run-time compares favorably with the run-time of $d^{poly(1/\epsilon}$ that is known to be best for the more challenging agnostic model. This is true for all values of bias $\alpha$, but is especially true when $\alpha$ is a small constant.
- The methods developed in this work seem potentially interesting in their own right.

**Weaknesses:**

- Not clear that the dependence on $\epsilon$ is best it can be.
- None of the algorithms run in fully polynomial time in all parameters.

**Questions:**

- I think there is a missing parenthesis in the end of Theorem 1.3
- Is it possible that for every small constant $c$, if $\alpha$ is promised to be at least $c$, then there is an algorithm running in time $\poly(d/\epsilon)$?
- Is it correct that for these halfspaces your algorithm runs in time $d^{O(1} 2^{\log(1/\epsilon)^{O(1)}}$?

**Limitations:**

Limitations are discussed adequately.

---

> ### Author Rebuttal · Authors · 2024-08-07
>
> We thank the reviewer for the time and effort in reviewing our paper and for the positive feedback. Below, we provide specific responses to the points and questions raised by the reviewer.
>
> > Is it possible that for every small constant $c$, if $\alpha$ is promised to be at least a constant, then there is an algorithm running in time $\mathrm{poly}(d/\epsilon)$?
>
> *Response:* Yes, it is possible in principle. Our lower bound does not rule out this possibility.
>
> > Is it correct that for these halfspaces, your algorithm runs in time $\mathrm{poly}(d)2^{\mathrm{polylog}(1/\epsilon)}$?
>
> *Response:* Yes, for any constant $\alpha$, our algorithm runs in $\mathrm{poly}(d)2^{\mathrm{polylog}(1/\epsilon)}$ time.
>
> > It is not clear that the dependence on $\epsilon$ is the best it can be.
> None of the algorithms run in fully polynomial time in all parameters.
>
> *Response:* We want to remark that although the dependence of the running time on $\epsilon$ might not be the best possible, it is still an exponential improvement over the previously best-known algorithm for the problem. Moreover, fully polynomial dependence on all parameters seems unlikely, given our SQ lower bound. Specifically, the $d^{\Omega(\log(1/\alpha))}$ dependence is inherent for SQ algorithms.

---

> > ### Comment · Reviewer_wFKn · 2024-08-07
> >
> > Thank you for your response. I think it is appropriate to update my rating from 6 to 7

---

### Author Rebuttal · Authors · 2024-08-07

We thank all reviewers for taking the time to read our manuscript carefully and for providing constructive and insightful feedback.

We provide detailed responses to each reviewer separately. We look forward to engaging in further discussion with the reviewers, answering questions, and discussing improvements.

---

### Decision · Program_Chairs · 2024-09-25

**Decision:**

Accept (spotlight)

**Comment:**

All reviewers agree that the paper studies an important problem, establishes sound results regarding the sample and computational complexity of distribution-dependent reliable learning, and introduces novel techniques that may find other applications in learning theory. In addition, a nearly matched statistical-query lower bound is established, showing the near-optimality of the proposed approach.

In more detail, authors observe that under the current noise model, it may be (information-theoretically) impossible to distinguish a candidate halfspace from the target one even though their L2-distance is bounded away. To resolve this fundamental issue, they propose a novel approach that performs random walk on a lower dimension and show that the random walk will hit a point at which the distance to the target halfspace is significantly decreased. Such techniques move away from prior works, which applied applied reduction to agnostic noise. As a result, the obtained sample and computational complexity are always lower than those of agnostic learning; in fact, for a wide regime, they are significantly lower.

The paper is very well written and easy to follow.